# An experimental investigation on the dark side of emotions and its aftereffects

**Lisette Ibanez[1]☯, Hayet Saadaoui[2]☯ ***

**1** UMR 5474/1135 CEE-M, INRA, Montpellier, France, **2** University of Sousse, Sousse, Tunisia

☯ These authors contributed equally to this work.
* hayet.saadaoui@iseahsb.u-kairouan.tn

**Data Availability Statement:** The database is available as Supporting Information S1 File.

**Funding:** Research Agency GREENGO research project (ANR-15-CE05-0008-01) to LI, https://anr.fr/en/. The funders had no role in study design,

## Abstract

The economic literature is so far overwhelmingly dedicated to the effect of incidental emotions on virtuous behavior. However, it is not so explicit for destructive behavior and the way it evolves with emotional states. To fill this gap, we explore how incidental emotions impact antisocial behavior in a laboratory experiment. As our vehicle of research, we used the open treatment of the joy-of-destruction mini-game. In addition to that, we elicited players' first and second-order beliefs via an incentivized questionnaire. We find that destructive behavior is driven by two motives: spite (Machiavellian traits) and preemptive retaliation (Expected destruction by partners). Emotional states do not impact destructive behavior directly. However, positive emotions brighten the expectations of other player beliefs on his partner's destruction, and indirectly reduces the willingness to destroy partner's money.

## 1. Introduction

Nowadays, choice behavior is certainly portrayed as the product of two motivational processes one more rational and deliberative, whereas the other more impulsive and driven by emotions and other motivational states [1, 2]. Despite the considerable interest shown by scholars to both processes, when it comes to modeling choice behavior like in the case of the Expected Utility Theory, one process is generally considered—the rational one. Abstaining from incorporating the other process—affect—in formal models is mainly explained by its unpredictable and complicated character as seen by many scholars.

However, in the last decades, many economists highlight the need to increase the explanatory and predictive power of such models by taking the affective sphere into consideration. This is largely attributed to behavioral economics. Since then, an enhanced understanding of the effect that emotions play in decision making is being investigated (see for example [3–7]. Emotions guide economic behavior in two different ways. First, some economists showed that expected emotions—people's predictions of what emotions they will experience after knowing the consequences of their own choice—can considerably influence decision making. Second, substantial research supports the idea that economic agents often refer to emotions experienced at the time of decision making i.e., immediate emotions. This type of emotions can be either connected to the decision at hand and provide individuals with systematic information

data collection and analysis, decision to publish, or preparation of the manuscript.

**Competing interests:** The authors have declared that no competing interests exist.

about their own values (integral emotions) or be unrelated to the decision at hand (incidental emotions).

In this study we will focus on incidental emotions and how they influence decision-making. While the impact of incidental emotion on decision making is well established [8, 9], its role is much more difficult to justify especially if we consider the fact that this specific type of emotions drives decisions in directions that run contrary to the predictions of consequentialist perspective.

The economic literature is so far overwhelmingly dedicated to incidental emotions' impact on virtuous behavior. Virtuous behavior is now well-established by economic researchers: individuals give money to their partners in dictator games [10, 11], place trust in trust games [12], reciprocate in gift exchanges [12, 13], punish free-riders in public goods games [14], and reject unfair offers in ultimatum games [15]. Thus, focusing on explaining the affective basis of altruistic, cooperative and fairness-minded homo reciprocans (as described by [16]), was an exciting task for behavioral economists who attempted to find explanations as to why individuals deviate from self-interested behavior and tend to be much more ethically motivated than assumed by economists. For instance, incidental emotions are found to influence prosocial behavior [17–23], trust [24–29], trustworthiness [20, 30, 31], and fairness [9, 32–42].

However, the literature is not so explicit for destructive behavior and the way it evolves with emotional states. Destructive behavior is observed in various domains: some people destroy public and private property like circulating malicious computer viruses, scratching someone else's car, puncturing tires, damaging buses and trams seats or throwing litter or cigarette buts into nature. Abbink and Sadrieh [43] argue that behavioral economists are "overstating the kindness of human nature". According to the authors, the focus on studying altruistic behavior and the neglect of people' opportunistic, greedy and nasty facet may create a selection bias.

Several authors investigate various determinants of destructive behavior using experimental games. Ranging from the Power-To-Take Game "PTTG" [44–46] and Money Burning Game [47–49] to the Joy-of-Destruction (JoD) Game [43, 50, 51], antisocial behavior is frequently attributed to an effort to reduce inequality, envy, fairness of inequality (morally questionable wealth sources), [52] self-protection, preemptive retaliation (when destruction is driven by the expectations of undergoing destruction [43]) or even pleasure. While there are many possible motivations for engaging in antisocial behavior, it seems appealing to assume a relationship between incidental emotions and destructive behavior. Indeed, in a fit of anger or in the grip of fear, actual decisions might be different from those anticipated in a cold state. For example, SARS-CoV-2 triggers fear and more selfish behavior, Black Live Matter demonstrations turn violent, . . . It is increasingly acknowledged that emotions represent a substantial factor for understanding and modeling economic behavior [53].

Despite the lack of experimental analysis on emotions and antisocial behavior, there are some studies that should be mentioned. It is to note that these studies mainly focus on integral emotions. In particular, most of the literature on the PTTG identifies negative emotions such as intense anger, irritation and contempt as the main driving force for punishing behavior. Basically, Bosman and van Winden [44] and different follow-up studies found that emotions were related to destruction decisions in a non-linear way. For instance, authors investigate the way emotions drive such behavior in relation to social ties [54], fairness perceptions and experienced emotions [55], waiting time [56], the demand for expressing emotions [57] and negative emotions [58]. In the same vein, using the JoD game, Caldara et al. [59] found that negative emotions, generated by the game, increase the destroyed amount. Based on the literature above, the mentioned games are responsible of triggering a range set of emotions susceptible of influencing spiteful behavior.

However, some external stimuli namely incidental emotions, i.e., emotions not related to the actual choice problem, could as well influence antisocial behavior. Fochmann et al. [60] show that positive (incidental) emotions reduce individuals' willingness to comply with social norms. Also, Blackwell and Diamond [61] provide evidence that a socially acceptable instrument for generating oxytocin (i.e., pleasant touch, or in this case, a hug) can reduce the spiteful behavior typically exhibited in the JoD game.

In this same perspective and in order to extend the literature on the JoD, which is relatively understudied, we aim to study how incidental emotions impact antisocial behavior in a laboratory experiment. As our vehicle of research, we use the open treatment of the joy-of-destruction mini-game (mini-JoD, see [50]). Participants play the mini-JoD game after having experienced a so-called emotion induction procedure that involves visual stimuli triggering negative, positive or neutral emotion. Each participant matched, randomly with another participant, has the possibility to destroy half of the endowment of his partner. The decision to destroy partner's money is costly, which means that the Nash equilibrium is to leave the partner's endowment unchanged.

Many specificities make the one-shot mini-JoD game appropriate to the research problem. First, it allows capturing the influence of incidental emotions on behavior since players' decisions are made simultaneously which exclude the possibility of aversive emotions (i.e., integral emotions) that players may feel in a sequential game such as the PTTG. Second, the use of an open (the player knows that destruction is inflicted by his partner) and costly (the player pays to destroy the endowment of his partner) game permits to investigate the hypothesis that, at least for some, destruction is intrinsically pleasurable as well as the susceptibility of incidental emotions to drive such motive.

In addition to that, we elicited players' first and second-order beliefs (beliefs about partners' actions, and, beliefs about partners' beliefs) on destruction behavior via an incentivized questionnaire. Geanakoplos et al. [62] argue that "in psychological games the payoff to each player depends not only on what every player does but also on what he thinks every player believes, and on what he thinks they believe others believe, and so on". In the context of destruction, Abbink and Herrmann [50] show that players' expectations of having their own endowments destroyed are positively correlated with the degree of destruction they are willing to inflict on others. Players do not want be the sucker under any circumstances and adopt destruction of partner's endowment as an act of preemptive retaliation [63].

Additionally, using a psychological personality inventory (the Short Dark Triad (SD3), we explore the impact of a dark trait of personality (i.e., Machiavellianism) on behavior in the JoD game. Actually, Machiavellianism has been associated with spitefulness [64, 65].

We contribute to the literature on antisocial behavior by reporting some results on what motivates destructive behavior in a JoD game. We show that choosing to destroy others' endowment is not directly influenced by emotional states. Nevertheless, emotions appear to have a particular role in the sense that they were found to impact expectations significantly. In addition, we found that preemptive retaliation and spitefulness are the main drivers of this behavior. Furthermore, we contribute to the literature on emotions and judgment and economic decision making. First, we add evidence on the role of incidental emotions in decision making, more specifically on non-virtuous behavior. Second, we induce incidental emotions using pictures. Several methods have been used to induce different emotional states such as music [66], a memory/imagination task [17] or movies [23].

The rest of the paper is organized as follows. Section 2 builds the theoretical foundation of our analysis and discusses the hypothesis to be examined and Section 3 describes the experimental design. The results are presented and discussed in Section 4. Section 5 concludes.

## 2. Theoretical framework and behavioral hypotheses

This section presents a formal model of spitefulness [67]. In this theoretical framework, individuals are faced to an immoral and economic costly decision and have to decide whether to destroy a part of their partner's monetary pay-off, knowing partners take a simultaneous decision, thus have the possibility to destroy their pay-off. Destruction of others' monetary pay-off implies a monetary cost.

We consider a one-shot two-players game, in which each player must decide simultaneously whether to reduce his partner's monetary pay-off partly (reduction parameter being equal to b), or to leave his pay-off unchanged. So, player $i$ chooses his destruction strategy $x_i \epsilon \{0,1\}$, $i \in \{1,2\}$. Each player forms beliefs on his partner's strategy. $\alpha_i^j$ represents the beliefs of player $i$ on partner $j$'s destruction behavior. Then an individual $i$'s utility can be written as follows

$$U_i = (1 - b\alpha_i^j)\pi_i + (S_i + R_i\alpha_i^j)x_i - cx_i$$

where $(1 - b\alpha_i^j)\pi_i$ represents the expected monetary pay-off, $(S_i + R_i\alpha_i^j)x_i$ the non-monetary reciprocity utility (Falk and Fehr, 2006) and $cx_i$ the monetary cost of destroying partner's monetary pay-off which we suppose to be linear.

The non-monetary utility of destruction has two components: $S_i$ which correspond to individual $i$'s satisfaction of destruction or spitefulness and $R_i\alpha_i^j$ the reciprocation term which represents the willingness to punish in case player $i$ expects partner $j$ to destroy his monetary pay-off. We consider the willingness to retaliate by $i$, $R_i$, to be determined by his second order beliefs, that is to say expected beliefs of player $j$ on $i$'s willingness to reciprocate, $R_i = \alpha_j^i$. In other words, $i$ anticipates the way player $j$ form beliefs on his own (i.e. $i$'s) retaliation behavior. This means that the higher the beliefs of player $j$ that player $i$ adopts destructive behavior, the higher the willingness to retaliate by $i$.

From the utility function, we can derive best response strategies for player:

$$x_i(\alpha_i^j) = \begin{cases} 0 & \text{if} \quad \alpha_i^j < (c - S_i)/\alpha_j^i \\ 1 & \text{if} \quad \alpha_i^j \geq (c - S_i)/\alpha_j^i \end{cases}$$

This means that player $i$ will decide to destroy his partners $j$'s pay-off when expectations of $j$ adopting destructive behavior towards himself, are high. This boundary is different for each player and decreases with player $i$'s spitefulness ($S_i$) and $j$'s expectations on $i$'s destruction behavior ($\alpha_j^i$).

And then equilibria decisions are represented in Fig 1.

Indeed, both spite and preemptive retaliation are found to be motivators for destroying the other's endowment [43, 63]. Players may just feel pleasure by harming others, or participants try to avoid being worse off than others under any circumstances by assessing the intention of the other player(s) and punishing players that intend to harm. Such behavior is in line with persistent findings in experimental economics suggesting that in numerous strategic environments individuals' preferences depend not only on the strategies played, and thus individual (pro-social) characteristic, but also on the beliefs they possess about other people's intentions and expectations [68]. We presume that this reasoning, as suggested in the theoretical framework, will be present in the player's mind when taking the decision to destroy or not the other's endowment in our experimental set-up. This allows us to dress our first hypothesis.

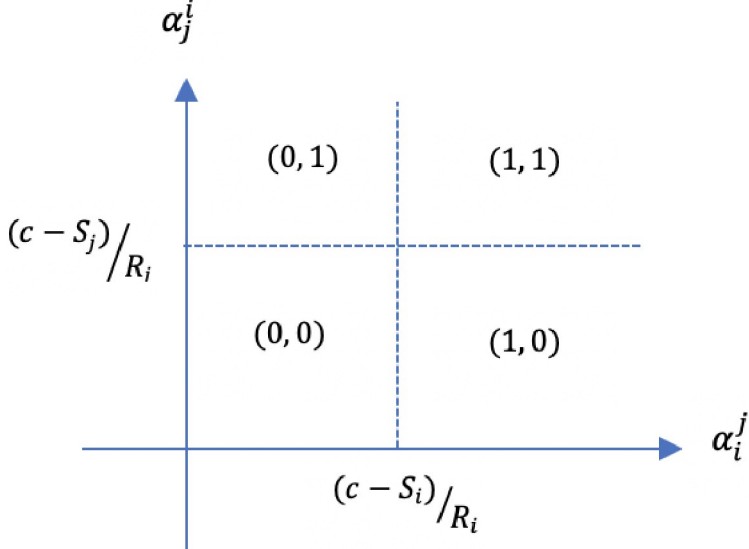

**Fig 1. Destruction decisions** $(x_i^*, \ x_j^*)$ **by both players i and j according first-order beliefs.**

**Hypothesis 1: The probability to adopt destructive behavior increases with the player i's spitefulness ($S_i$), his/her expectations on partner j's willingness to destroy ($\alpha_i^j$), and partner j's beliefs on player i's willingness to retaliate ($\alpha_j^i$).**

The formal framework presented above is based on rational decision making and does not consider the emotional state of players when they have to choose the part of their partner's monetary pay-off to destroy. However, the recent economics literature supports the evidence that incidental emotions can have a substantial impact on decision making [69]. To discover its impact, many studies have manipulated the affective state by inducing different valences and arousal levels with, among others, music, images and videos. More recently, Drouvelis and Grosskopf [23] use movies and show that angry individuals adopt less prosocial behavior than happy individuals. Similarly, Blackwell and Diamond [61] found that the act of hugging acts as a positive social stimulus that causes a hormonal generation of empathy which appears to reduce antisocial behavior. Furthermore, half as many participants decided to destroy a portion of an anonymous participant's endowment when they hugged before playing the JoD game.

Frijda [70] indicates that incidental frustration makes players angry with anyone, regardless of the source of frustration: I hit my head on the kitchen shelf and so I punish you; I hit my thumb with a hammer and so I punish you. The most plausible explanation stems from the relationship between incidental emotions and particular judgmental tasks, such as attributing responsibilities and blame, as well as the inclinations associated with these attributions. Feigenson [71] indicates that incidental emotions affect attributions of responsibilities and blame through two judgmental paths. The first path concerns the way incidental emotions drive individuals' processing of available information about the target of their judgement and is explained by the appraisal tendency process. This process shows that even when individuals are aware that the source of their emotional state is independent of the judgement target, the emotion continue to affect their judgements (consciously or not). For instance, research consistently predicts that people who are angry blame more. Card and Dahl [72] and Munyo and Rossi [73] suggest that an unexpected loss suffered by a local soccer team is associated with

substantial increases in domestic violence and violent crime. Similarly, Keltner et al. [74] suggest that anger and sadness influenced participants' attributions of causal responsibility, which in turn affect blaming. The second path concerns how people take the experienced emotion as informative about the judgement target and is explained by the affect-as-information model. While the experienced emotion is provoked by stimuli not relevant to the judgement task, it is regarded as directly informative of the target. Indeed, people seem to misattribute their emotional experience to the target of judgement instead of its exact source. For example, on rainy days, people gauge their life satisfaction more negatively [75]. Moreover, emotional states may shape expectations. Fochmann et al. [60] provide evidence that individuals have a lower willingness to comply with social norms and fraud more after being primed with positive incidental emotions compared with aversive emotions. Indeed, being positively mooded, may reduce expectations to be audited.

Building on these insights, and considering emotions as an exogenous shock, we derive our second behavioral hypothesis.

**Hypothesis 2: Under a positive emotional state (respectively negative emotional state), an individual will destroy less (respectively more) than under a neutral emotional state, as**

a. **His spitefulness will be tempered (resp. reinforced) i.e.**
$S_{i,\text{positive emotion}} < S_{i,\text{neutral emotion}}$ and $S_{i,\text{negative emotion}} > S_{i,\text{neutral emotion}}$

b. **His expectations on partner j's destructive behavior will be reduced (resp. increased), i.e.** $\alpha^{j}_{i,\text{positive emotion}} < \alpha^{j}_{i,\text{neutral emotion}}$ and $\alpha^{j}_{i,\text{negative emotion}} > \alpha^{j}_{i,\text{neutral emotion}}$

c. **His expectations on partner j's beliefs on player i's willingness to retaliate decreases (resp. increases), i.e.** $\alpha^{i}_{j,\text{positive emotion}} < \alpha^{i}_{j,\text{neutral emotion}}$ and $\alpha^{i}_{j,\text{negative emotion}} > \alpha^{i}_{j,\text{neutral emotion}}$

Moreover, we expect players' first-order beliefs to be contingent on their second-order beliefs, i.e., their expectations of what others expect of them. This means that participants' destructive behavior is not only motivated by their expectations about their counterparts' behavior (first-order beliefs), but also depends on their second-order beliefs. Interestingly, some recent research investigates on the way expectations influence actual behavior. For instance, if one expects anger to be a good driver for performance, behavior will actually become better performing [76]. This leads us to our third hypothesis.

**Hypothesis 3: Participants' expectations on partners' beliefs that one will retaliate (second-order beliefs) will increase destruction through expectations on partners' destruction behavior (first-order beliefs).**

## 3. Experimental design

A computerized experiment was conducted between June and October 2018 at the Montpellier Laboratory for Experimental Economics (LEEM). In total, 172 participants (69 women; average age 24.48, SD 6.730) were recruited randomly from the LEEM database (via the ORSEE software). 9 out of 10 subjects had previously participated in a laboratory experiment. We ensured, however, that none had done an experiment with similar parameters. Participants were randomly assigned to a cubicle workstation and told that the experiment was composed of three parts, but that they only received instructions for each part one at a time (after completing a previous part) and that only one, randomly chosen, was to be paid at the end. We run 9 sessions, 3 for each treatment: a control treatment with a neutral emotion induction; a positive emotion treatment and a negative emotion treatment. We ran an ex-post power analysis using the G*power software which revealed that our sample size (respectively 55, 57 and 60

**Table 1. The experimental protocol.**

| | |
|---|---|
| Task 1 | Emotion induction and assessment |
| Task 2a | The joy of destruction mini-game |
| Task 2b | Elicitation of 1st order and 2nd order beliefs |
| Task 3 | Social Value Orientation game |
| Task 4 | Ambiguity and risk preferences game |
| Task 5 | Dark Triad & Socio-demographic questionnaires |

observations for the three treatments) is sufficient (power = 0.8) to observe an effect size of 0.4. While Cohen [77] recommends that wherein d = 0.20 indicates a small effect, d = 0.50 indicates a medium effect and d = 0.80 indicates a large effect, Dancey & Reidy [78] specify that small and medium effects are more willingly available when evaluating behavioral and psychological constructs as it is the case of constructs investigated herein.

We consider an in-between design. In each treatment, participants are matched in pairs but they never learn the identity of the person to whom they are matched with, and instructions were as neutral as possible (for instance, the word "reduce" was used, with no mention of the term "destroy"). An experimental currency was used, the "ECU", convertible in French euros at the end of the experiment at the rate of 15 cents per ECU. On average, the participants received 14 euros (including the show-up fee) for approximately 45 min of participation. Earnings were paid in cash, privately, at the end of the session.

The protocol of the experiment is described in Table 1. Tasks 2, 3 and 4 were incentivized. However, at the end of the experiment only one of three tasks was randomly chosen to be actually paid to the participants. More details are provided on the different tasks in section 4.

## 4. Materials and methods

### 4.1 Incidental emotion manipulation

In Task 1, we induce positive and negative emotions by exposing subjects to slideshows of 15 pictures selected from the Geneva Affective Picture Database (GAPED); see [79]. A control treatment was run as well where 15 neutral pictures were shown to participants. Emotional states are measured by using the "affective slider" method suggested by Betella and Verschure [80]. Participants choose a value in between 0 and 99 that reflects best their affective states by moving a mouse pointer over a slider. They have to pick a value for their degree of pleasure (valence) and their degree of excitement (arousal). For valence, the leftmost position represented the most negative value (i.e., very sad feeling), and the rightmost position represented the most positive value (i.e., very happy feeling). For arousal, the leftmost position means that the subject is very calm, and the rightmost position means that the subject is very excited. The emotional state of the participants was manipulated at two points during the experiment. A first assessment was made upon arrival and a second assessment was made after the induction of emotions (slideshow of 15 pictures).

### 4.2 The joy of destruction mini-game, and Elicitation of first order and second order beliefs

In Task 2a, directly after the emotional state induction, the players participate in the one-shot mini-JoD game [43]. Each participant was matched randomly with another participant (a pair). For each pair, the two players (partners) were exposed to the same type of emotion and were endowed with 100 ECUs each, and both players simultaneously decided whether to pay 10 ECUs in order to decrease the counterpart's payoff by 50 ECUs (destructive choice) or

whether to keep the payoffs unchanged (non-destructive choice). We consider the open treatment, i.e., the destruction is ex-post perfectly observable.

In order to distinguish reciprocal motivations from spite, an incentivized post-experimental questionnaire composed of two questions (Task 2b) was implemented allowing to collect 1st order and 2nd order beliefs. Each correct guess was rewarded with 20 ECUs.

We suppose 1st order beliefs (i.e., expected partners' destruction), to be a proxy for α, i.e., player's expectations on partner's destructive behavior. To elicit 1st order beliefs, we asked each participant to indicate how many persons in the same experimental session, in the same role as his partner, will choose to reduce their partners' endowments.

Then, the participant was asked to state his 2nd order beliefs (i.e., expected partners' beliefs on destruction of other players), by indicating the guess of his partner on the number of persons, in the same role as himself, having chosen to reduce their partners' endowment. We suppose second order beliefs to be a proxy for R, i.e., player's willingness to reciprocate.

### 4.3 Control variables

We established a proxy for spite (S) by measuring the dark trait (Machiavellianism), with the 9-item Short Dark Triad (SD3) [81] in Task 5. Participants were asked how much they agreed (1 = not at all; 5 = extremely) with statements such as: "Make sure your plans benefit yourself, not others". This is a non-clinical measure of Machiavellianism, allowing the evaluation of empirical associations in normal populations.

Because social preferences are of crucial importance in understanding interdependent decision-making behavior [82], we wish to explore their influence in our experiment (Task 3). For this reason, we elicit subjects' social value using the social value orientation (SVO) slider measure developed by Murphy et al. [83]. Social preferences based on the SVO framework have been found to be predictive to behavior in many games (e.g., [84]). For instance, some studies show a positive association between individuals' general level of pro-sociality and their willingness to engage in inter-group conflict [85].

In Task 4, we elicited risk and ambiguity aversion using Ellsberg urns [86] since several forms of antisocial behavior are found to be linked to decisions made under conditions of ambiguity and risk [87]. The choice of this specific procedure is justified by its ability to test both ambiguity and risk at the same time contrarily to other procedures.

Lastly, at the end of the experiment, we collected various socio-demographic information: gender, age, level of education, and whether participants already participated to a lab experiment (= "Experience" variable).

All variables elicited during the experiment are described in Table 2.

## 5. Results

In this section, we explore whether, and if so, how incidental emotions, both positive and negative, influence behavior in the Joy of Destruction mini-game. We investigate to what extent destruction is motivated by spitefulness (Machiavellianism), expectations on partners' willingness to destroy (Expected partner's destruction) and willingness to reciprocate (Expected partners' belief on destruction), and the way the incidental emotions impact these motivations. To analyze our results, we use non-parametric tests and models as Skewness and Kurtosis test for normality shows that residuals are not normally distributed ($p < 0.5$).

### 5.1 Manipulation check

We start by examining the effectiveness of our emotion induction, i.e., whether in the positive emotion (respectively negative emotion) treatment participants feel happy (respectively sad).

**Table 2. Descriptive statistics of variables.**

| Variables | Definition | Mean | S. D | Min | Max |
|---|---|---|---|---|---|
| Destruction | A binary variable that takes the value of one if the subject X chooses to destroy his partner Y's endowment and zero otherwise. | 0.104 | 0.306 | 0 | 1 |
| Positive emotions | A binary variable that corresponds to one if positive emotion is induced and to zero otherwise | 0.348 | 0.477 | 0 | 1 |
| Negative emotions | A binary variable that corresponds to one if negative emotion is induced and to zero otherwise | 0.319 | 0.467 | 0 | 1 |
| Expected partner's behavior | The expected number of players Y out of 10 (including the player X's partner) who have chosen to reduce their partner's income in decision 1. | 2.284 | 2.450 | 0 | 10 |
| Expected partner's beliefs | The expected number of players X out of 10, player X expects player Y to belief to destroy their partner Y's income in decision 1 | 2.668 | 2.673 | 0 | 10 |
| Machiavellianism | The overall mean of the nine items. | 2.737 | 0.596 | 1.333 | 5 |
| Prosocial | A binary variable that takes the value of one if the subject is prosocial and zero otherwise. | 0.5 | 0.501 | 0 | 1 |
| Risk Averse | A binary variable that takes the value of one if the subject is risk averse (prefer a certain payoff to a lottery with known probabilities) and zero otherwise. | 0.726 | 0.446 | 0 | 1 |
| Ambiguity Averse | A binary variable that takes the value of one if the subject is ambiguity averse (avoid options whose outcome probabilities are unknown) and zero otherwise. | 0.848 | 0.359 | 0 | 1 |
| Gender | A binary variable that takes the value of one if the subject is male and zero otherwise. | 0.598 | 0.491 | 0 | 1 |
| Experience | A binary variable that takes the value of one if the subject has already participated to a lab experiment and zero otherwise. | 0.895 | 0.306 | 0 | 1 |

We measure the emotional state by valence, and control for similar arousal. In Table 3, we show that after the slideshow viewing, participants feel happier (respectively adder) in the positive emotion treatment (respectively negative emotion one). We don't observe a significant difference in arousal for the three treatments. We also compare the emotional state before and after the slideshow of pictures for both valence and arousal, to make sure that the difference observed is not due to the emotional state at arrival. As shown in Table 3, the equality of population tests after emotion induction are rejected both for valence and arousal. A more in-depth analysis (not presented in the table but available upon request) shows that participants experience less pleasure in the negative emotion treatment (Tukey HSD post hoc test, p< 0.001) and the neutral emotion one (Tukey HSD post hoc test, p<0.02) than in the positive emotion treatment. Also, differences occur between the negative condition and the neutral condition (Tukey HSD post hoc test, p<0.001). Furthermore, we observe a slightly higher intensity in which the emotion is experienced for both the positive and negative emotion in comparison with the neutral emotion, however difference of the arousal level before and after the pictures shown are not significantly different between the three treatments (Tukey HSD post hoc test,

**Table 3. Mean of emotional states after slideshow viewing, and the difference in the emotional state before and after emotion induction, for valence and for arousal (S.D. in brackets).**

| Emotion | Nb of obs. | Valence after slideshow | Arousal after slideshow | Difference for Valence | Difference for Arousal |
|---|---|---|---|---|---|
| Positive | 60 | 84.45 (19.85) | 72.95 (28.21) | 2.75 (8.1) | 0.68 (13.47) |
| Negative | 55 | 51.51 (28.56) | 76.78 (21.73) | -25.28 (25.67) | 1.62 (24.76) |
| Neutral | 57 | 68.24 (23.13) | 69.33 (28.59) | -6.11 (15.25) | -4.11 (18.9) |
| Kruskal-Wallis equality of population test (p-value) | | 0.0001*** | 0.480 | 0.0001*** | 0.098* |

Significant levels

\*** p<0.01

\** p<0.05

\* p<0.1

**Table 4. Treatment-specific summary statistics (S.D. in brackets).**

| | Nb of Obs | % of destruction (S.D) | | | Machiavellian traits |
|---|---|---|---|---|---|
| | | Destruction behavior, % | Expectations on partners' destruction, % | Expectations on partners' beliefs on destruction, % | |
| Overall | 172 | 10.46% (0.31) | 22.85% (2.45) | 26.69% (2.79) | 2.74 (0.6) |
| *Emotion* | | | | | |
| Positive | 60 | 8.33% (0.28) | 19.67% (2.56) | 22.17% (2.79) | 2.77 (0.58) |
| Negative | 57 | 10.90% (0.33) | 22.18% (2.45) | 26.36% (2.47) | 2.74 (0.58) |
| Neutral | 55 | 12.28% (0.31) | 26.8% (2.31) | 31.75% (2.69) | 2.71 (0.64) |
| Kruskal-Wallis equality of population test *(p-value)* | | 0.78 | **0.096**<sup>*</sup> | 0.13 | 0.56 |

Standard deviations in brackets

*p < .1

**p < .05 and

***p < .01

p>0.1). In summary, these results provide clear evidence that our manipulation to induce positive (respectively negative) emotional states has been successful.

## 5.2 The influence of emotions on destructive behavior

We start by examining the percentage of the monetary endowment destroyed in the one-shot joy-of-destruction mini-game. Our finding is consistent with Abbink and Herrmann's findings [50], about one in nine subjects (10.46%) exhibits spiteful behavior and destroys his anonymous partner's monetary endowment at own costs. Results regarding destructive behavior across treatments are reported in Table 4.

We observe a lower destruction level in the positive emotion treatment (8.33%) as compared to the negative emotion treatment (10.90%) and the neutral emotion treatment (12.28%). However, there is no significant difference between the three treatments (p-value = 0.78).

We use a Mann-Whitney test to perform binary treatment comparisons (Table 5), and show that the hypothesis of no difference across treatments for destruction cannot be rejected.

**Table 5. Binary treatment comparisons (Wilcoxon rank-sum Mann-Whitney test, z-statistic).**

| | Destruction behavior | Expectations on partners' destruction behavior | Expectations on partners' beliefs on destruction behavior | Machiavellian[2] traits |
|---|---|---|---|---|
| **Positive vs Negative** | -0.47 (0.64) | -1.05 (0.29) | -1.08 (0.28) | 1.02 (0.31) |
| **Positive vs Neutral** | -0.70 (0.48) | **-2.09**<sup>**</sup> (0.037) | **-1.99**<sup>**</sup> (0.046) | -0.5 (0.62) |
| **Negative vs Neutral** | -0.23 (0.82) | -1.25 (0.21) | -0.94 (0.35) | -0.67 (0.5) |

Standard deviations in brackets

*p < .1

**p < .05 and

***p < .01

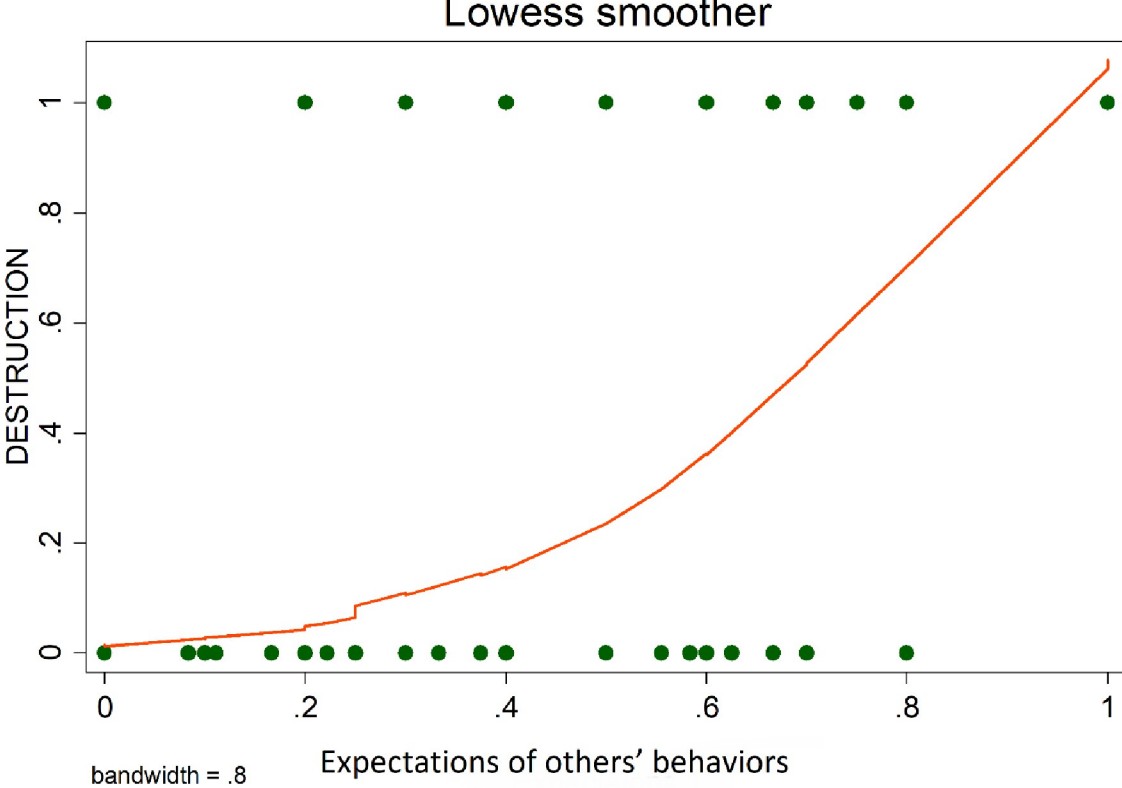

**Fig 2. Expectations of others' behaviors & destruction.**

**Result 1: Destruction levels do not statistically differ between different emotional states.**

### 5.3 The influence of emotions on players' first and second-order beliefs, and Machiavellian traits

Table 4 also provides information about the effect of incidental emotions on the expectations subjects have about other players' behaviors and beliefs, as well as on their Machiavellian traits. We notice that, under a positive emotional state, individuals expect less destruction from their partners (19,67%) than in the negative (22.18%) and neutral (26.8%) emotional states. A Mann-Whitney test (Table 5) gives further support that individuals' expectations about other players' behavior, in the positive and neutral conditions, are significantly different (z-statistic = -2.09, p<0.05). Moreover, results suggest that, when induced with positive emotions, individuals expect others to think positively about them and to not expect them to exhibit destructive behaviors (z-statistic = -1.99, p = 0.046).

**Result 2: Compared to neutral emotions, positive emotions lower individual's expectations on partners' destructive behavior and beliefs.**

In addition, Table 5 also shows that a negative emotional state doesn't influence significantly expectations, and that emotions don't impact Machiavellian traits.

### 5.4 Expectations and destruction

To present the underlying nature of the relationship between destruction and expectations, Lowess smoother curves were used. In line with the literature (e.g., [50]), we observe that

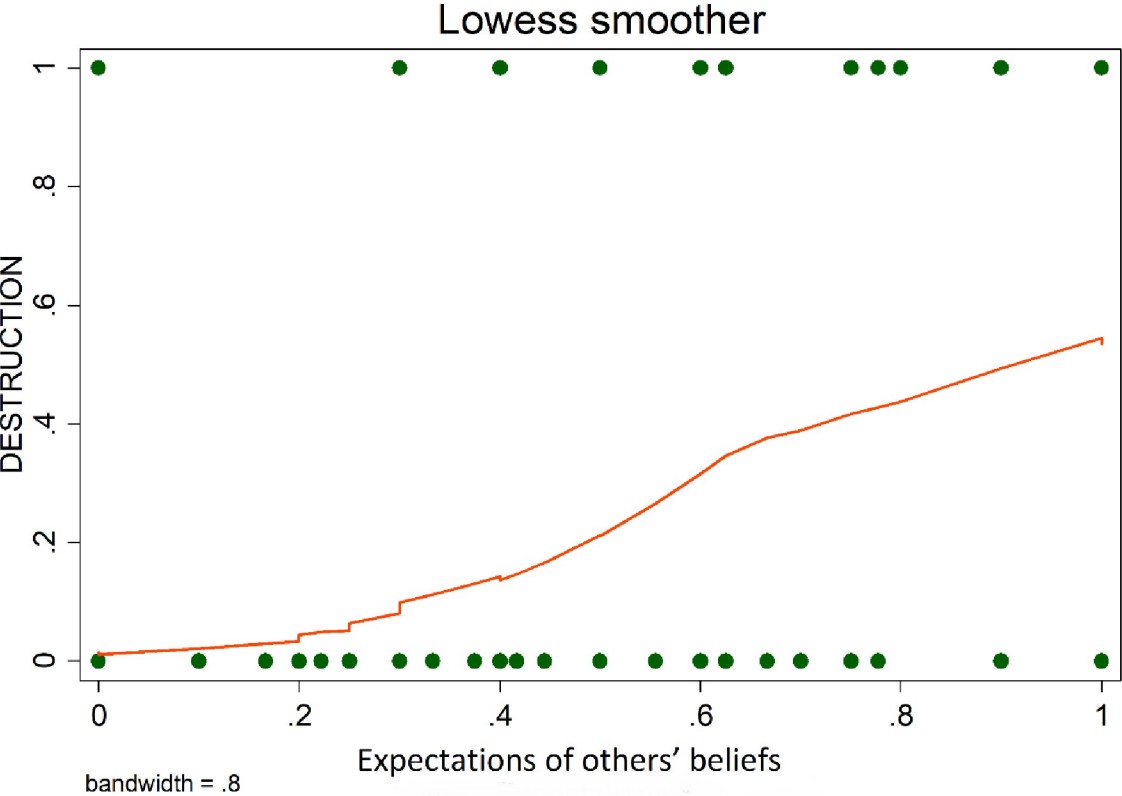

**Fig 3. Expectations of others' beliefs & destruction.**

players' expectations of having their own endowments destroyed are positively correlated with the degree of destruction they are willing to inflict on others (see Fig 2). Also, as shown by Fig 3, the more individuals suppose that other players expect them to destroy their endowments, the more they destroy.

## 5.5 Determinants of destructive behavior

We perform some econometric analyses, using STATA software, to better understand the drivers of destructive behavior and the role of incidental emotions. We first implement logistic regression analyses to understand what determines destruction behavior directly, and in what extent emotions influence the willingness to destroy partner's endowment. Second, we use Tobit regressions to analyze indirect effects of expectations on destruction as the dependent variables are left and right censored.

A logistic regression analysis is used to examine the impact of our three behavioral variables (i.e., expectations on partners' destruction, expectations on partners' beliefs, and Machiavellian traits) on our dichotomous dependent variable which is participant's decision to destroy a part of his partners endowment. We checked for collinearity of independent variables, and made sure that all variables could be maintained in the regression (variance inflation factor < 4). In Table 6, we present results and we show that destructive behavior is mainly driven by spitefulness (Machiavellianism) and expectations about partner's destructive behavior (Expected partners' destruction). More precisely, we observe in Model 1 that an increase in expectations on partners' destruction behavior, i.e., if one expects one extra partner out of ten to adopt

**Table 6. Average marginal effects of logit regressions for destruction behavior.**

| Dependent variable Destruction | Model 1 | Model 2 | Model 3 | Model 4 |
|---|---|---|---|---|
| *Independent variables* | AME (SD) | AME (SD) | AME (SD) | AME (SD) |
| Expected partners' destruction | **0.048**\*\*\* (0.014) | **0.05**\*\*\* (0.014) | **0.053**\*\*\* (0.015) | **0.051**\*\*\* (0.016) |
| Expected partners' beliefs on destruction | 0.002 (0.013) | 0.0016 (0.013) | 0.001 (0.012)) | -0.001 (0.013) |
| Machiavellianism | **0.091**\*\* (0.036) | **0.092**\*\* (0.036) | **0.116**\*\* (0.041) | **0.122**\*\* (0.044) |
| Positive emotion | | -0.033 (0.042) | -0.018 (0.038) | -0.017 (0.036) |
| Negative emotion | | -0.013 (0.046) | -0.003 (0.044) | -0.006 (0.044) |
| Positive emotion* Machiavellianism | | | **0.207**\*\*\* (0.059) | **0.211**\*\*\* (0.065) |
| Negative emotion* Machiavellianism | | | 0.057 (0.057) | 0.056 (0.063) |
| *Control variables* | | | | |
| Risk Averse | | | | -0.056 (0.043) |
| Ambiguity Averse | | | | -0.027 (0.038) |
| Prosocial | | | | -0.05 (0.036) |
| Gender | | | | -0.015 (0.037) |
| Experience | | | | 0.018 (0.061) |
| Log Likelihood | -32.89 | -32.59 | -29.28 | -27.63 |
| LR Chi2 | **49.53**\*\*\* | **50.13**\*\*\* | **56.75**\*\*\* | **60.04**\*\*\* |
| Pseudo R2 | 0.43 | 0.43 | 0.49 | 0.52 |
| Number of observations | 172 | 172 | 172 | 172 |

\*p < .1

\*\*p<0.05

\*\*\*p<0.01; SD: standard deviation; AME: average marginal effect

destruction behavior, the probability to adopt destructive behavior increases with point 0.048. Similarly, an increase in the score of the Machiavellian trait increases the probability to destroy partners endowment with point 0.091. Expectations on partner's beliefs which is considered to be a proxy for willingness to reciprocate don't determine directly destruction. We thus only partly provide evidence for hypothesis 1.

**Result 3: The probability to adopt destructive behavior increases with the player's Machiavellian traits and his/her expectations on partner's destruction behavior.**

Thus, overall findings show that Machiavellianism is found to be a strong motive for destruction. This result is expected since people who possess Machiavellian traits were already shown to be manipulative [88], opportunistic [89] and unethical [90], because these individuals are convinced that the ends justify the means, they neglect social interests [91].

Moreover, expectations about others' behavior are also a good predictor for destructive behavior. In other words, individuals who expect their partners to destroy their money respond to this expectation by destroying their partners' money which is consistent with previous researches (e.g., [43]). Both results are in line with Hypothesis 1.

We don't observe a direct effect of emotions on destruction. However, positive emotions act as a mediator on Machiavellianism. More specifically, we observe that participants, under a positive emotional state, have a higher probability to adopt destruction behavior the higher their Machiavellian traits. So, result 4 only confirms partly hypothesis 2a.

**Result 4: Positive emotions increase destruction for individuals with high traits of Machiavellianism.**

One possible explanation for this result could be found in the empirical studies that have revealed the darker side of happiness. In this vein, Gruber et al. [92] indicates that experiencing happiness is not always positive. People who are in the pursuit of happiness tend to be to be more depressed, miserable, and unhappy. According to Tan and Forgas [18] there exists a positive and significant relationship between happiness and selfishness. Further, Tamir and Bigman [93] demonstrate that positive (respectively negative) emotions can have a negative (respectively positive) impact on people. Thus, there exists evidence that positive emotion might lead to counter-intuitive effects. Adding to this, previous research has also shown that individuals with high levels of Machiavellianism usually endorse ego-centric or antisocial notions of well-being where they tend to prioritize their personal happiness over other people's happiness [94, 95].

Looking in more detail to the indirect impact of expected partners' destruction and beliefs on destruction, we start by regressing first order beliefs on the second order beliefs (Table 7). To do so, we use censored Tobit regressions (we have 65 left-censored observations and 2 right-censored ones).

We find a significant and positive impact of $2^{nd}$ order beliefs on $1^{st}$ order beliefs. This means that expectations on partners' destruction and partners' beliefs on destruction are positively correlated. More precisely, an 10% increase in expected partners' beliefs (i.e., participant expects his/her partner to belief that one extra participant out of 10 will destroy his/her partner's endowment), will increase expected partners' destruction with around 10.3%. In other words, the relation is almost unit elastic.

At this point, we observe that expectations about others' behavior are found to be strong predictor of destruction (Table 6) and these expectations were also found to be strongly correlated with expectations about others' beliefs (Table 7). These results show that there exists on indirect effect of expectations about others' beliefs on destruction, and thus confirms hypothesis 3. In other words, we provide evidence for the existence of recursive preemptive retaliation: besides the fact that destruction increases when expectations on partner's destructive behavior increases, expectations on partner's destructive behavior increases when expectations on partner's beliefs increases.

### Result 5: Participants' second-order beliefs increase destruction through first-order beliefs.

Regarding the impact of emotion, we observe that emotions don't impact expectations on partners' destruction behavior directly. However, positive emotions play a role in the way participants form expectations. For individuals with low expectations on partners' beliefs, positive emotions lower their expectations on partners' destructive behavior. For instance, in Model 6, we observe that under a positive emotion, a participant who expects partners to belief that others will not destroy any money, will have lower expectations on partner's destructive behavior (decrease of 1.19 point). On the other hand, for individuals with high expectations on partners' beliefs, expectations that partners decide to destroy increases. Indeed, we observe an increase of 1.725 for participants who expect their partners to belief that all other participants, i.e., 10 out of ten, will adopt destruction behavior. This result only confirms partly Hypothesis 2b, but seem to indicate that positive emotions influence expectations on destruction and thus indirectly influence destructive behavior.

As explained in a previous point, positive emotions can have a negative impact on individuals' thoughts and behaviors. It is worthy to mention that a growing body of research investigating the relationship between emotional experience and social information processing indicates that when people are performing a social judgment task the type of strategy they use may be affected by their momentary emotional state (for a review, see for example [96]). In the same

**Table 7. Average marginal effects of tobit regressions for expected partners' destruction.**

| Dependent variable Expected partners' destruction | | Model 5 | | Model 6 | | Model 7 | |
|---|---|---|---|---|---|---|---|
| *Independent variables* | | AME | SD | AME | SD | AME | SD |
| Expected partners' belief on destruction | | **1.031**\*\*\* | 0.06 | **1.03**\*\*\* | 0.058 | **1.032**\*\*\* | 0.059 |
| Machiavellianism | | 0.168 | 0.245 | 0.156 | 0.245 | 0.054 | 0.252 |
| Positive emotion | | -0.232 | 0.349 | -0.412 | 0.361 | -0.335 | 0.361 |
| Negative emotion | | -0.14 | 0.341 | -0.11 | 0.345 | -0.11 | 0.344 |
| Positive emotion* expected partners' belief (= nb of destructors) | 0 | | | **-1.19**\*\* | 0.561 | **-1.11**\*\* | 0.563 |
| | 1 | | | **-0.9**\* | 0.466 | **-0.821**\* | 0.468 |
| | 2 | | | -0.607 | 0.392 | -0.53 | 0.393 |
| | 3 | | | -0.315 | 0.352 | -0.238 | 0.352 |
| | 4 | | | -0.024 | 0.358 | 0.053 | 0.357 |
| | 5 | | | 0.268 | 0.409 | 0.345 | 0.405 |
| | 6 | | | 0.56 | 0.489 | 0.637 | 0.484 |
| | 7 | | | 0.851 | 0.588 | 0.928 | 0.582 |
| | 8 | | | **1.142**\* | 0.7 | **1.22**\* | 0.691 |
| | 9 | | | **1.434**\* | 0.813 | **1.512**\* | 0.805 |
| | 10 | | | **1.725**\* | 0.933 | **1.8**\*\* | 0.924 |
| Negative emotion* expected partners' belief (= nb of destructors) | 0 | | | -0.326 | 0.556 | -0.355 | 0.557 |
| | 1 | | | -0.245 | 0.455 | -0.263 | 0.456 |
| | 2 | | | -0.164 | 0.377 | -0.17 | 0.377 |
| | 3 | | | -0.083 | 0.337 | -0.078 | 0.336 |
| | 4 | | | -0.002 | 0.35 | 0.015 | 0.347 |
| | 5 | | | 0.079 | 0.409 | 0.107 | 0.405 |
| | 6 | | | 0.160 | 0.5 | 0.2 | 0.495 |
| | 7 | | | 0.242 | 0.606 | 0.292 | 0.601 |
| | 8 | | | 0.323 | 0.723 | 0.385 | 0.718 |
| | 9 | | | 0.404 | 0.845 | 0.477 | 0.840 |
| | 10 | | | 0.485 | 0.971 | 0.57 | 0.966 |
| *Control variables* | | | | | | | |
| Risk Averse | | | | | | -0.015 | 0.327 |
| Ambiguity Averse | | | | | | 0.248 | 0.298 |
| Prosocial | | | | | | 0.062 | 0.288 |
| Gender | | | | | | -0.154 | 0.298 |
| Experience | | | | | | **-0.842**\* | 0.444 |
| **Log Likelihood** | | -238.14 | | -235.47 | | -232.99 | |
| **LR Chi2** | | **210.07**\*\*\* | | **215.42**\*\*\* | | **220.38**\*\*\* | |
| **Pseudo R2** | | 0.31 | | 0.314 | | 0.32 | |
| **Number of observations** | | 172 | | 172 | | 172 | |

\*p < .1

\*\*p<0.05

\*\*\*p<0.01; SD: standard deviation; AME: average marginal effect

vein, certain positive emotions lead people to rely more on highly accessible cognitions, such as beliefs, expectations, and stereotypes (e.g., [97]). Importantly, happiness has been usually associated with the use of more superficial or cursory styles of thinking. For instance, some studies on mood and persuasion (e.g., [98]) support this evidence and document that prior to the presentation of a persuasive message, happy people are less affected by variations in

argument quality. Indeed, happy people often prefer to base their reactions more on simple cues [49]. Thus, we assume that, under a positive emotional state, individuals have adopted a superficial thinking that disabled them from engaging in close scrutiny of the situation. It appears that these individuals expected their partner's destruction to increase because they have interpreted the suspicious situation as a threat to be handled with precaution.

It is to notice that participants having already participated to a lab experiment expect less destruction behavior by partners. An interpretation of this result might be simply that because of their experience, their expectations are closer to real behaviors in the lab. Nevertheless, this interpretation might be debatable, and one might also argue that experienced participants have learnt specific behavior in previous experiments and thus are more prone to detect the goal of the experiment which might bias decisions.

A Tobit regression (with 66 left-censored and 3 right-censored observations) explaining second order beliefs (Table 8), shows again a positive link between expectations on partners' destruction behavior and expectations on partners' beliefs. A 10% increase in expectations on partners' destruction behavior (i.e., One expects one extra partner out of 10 to adopt destruction) increases expectations on beliefs by around 12%.

We also observe that positive emotions reduce significantly expectations on partners' beliefs (For instance, in model 9, expectations on partners' belief are reduced by -0.739). Again, positive emotions act as a catalyst. Participants under a positive emotion who have low expectation on partners' destruction, will have lower expectations on partners' beliefs than under a neutral emotion. To illustrate, we see that in model 9, if one expects 0 partners out of ten to destroy, the expected partners' belief will be reduced by 1.36 (i.e., which represents more than one person).

To resume Tables 6–8, as second order beliefs are a good predictor for first order beliefs, and first order beliefs a good predictor for destructive behavior, we conclude that positive emotions indirectly influence destruction.

**Result 6: Positive emotions reduce destruction through second order beliefs.**

At last, in terms of participants specific characteristics, we observe that participants having already participated to a lab experiment expect partners to belief that others adopt more destructive behavior than expected by participants without any experience on lab experiments. We also observe that men have different expectations on partners' beliefs on one's own destruction as women: men expect partners to belief them to be less destructive. This result might be explained by findings that men are found to be more optimistic than women [99].

## 6. Discussion et conclusion

The main objective of the current study was to examine drivers for destructive behavior and whether incidental emotions are susceptible to influence people's choice to costly destroy others' endowment. In addition, we were interested in capturing the mechanism underlying first and second-orders beliefs and destruction.

A first finding highlights that expectations about partners' destructive behavior and beliefs represent a strong predictor of one own's destructive behaviors. This result is valuable insofar as it reveals a possible explanation as why people's expectations about others' destructive behavior, in the context of JoD, are always negative. Actually, people hold these expectations because they expect others to think the same way they do themselves. In addition to the positive influence of first order beliefs on the decision to destroy which is well-established by the literature and confirmed by our research, second-order beliefs were found to impact destruction through their impact on first-order beliefs.

**Table 8. Average marginal effects of tobit regression for expected partners' beliefs on destruction.**

| Dependent variable Expected partners' belief on destruction | | Model 8 | | Model 9 | | Model 10 | |
|---|---|---|---|---|---|---|---|
| *Independent variables* | | AME | SD | AME | SD | AME | SD |
| Expected partners' destruction | | **1.196**\*\*\* | 0.071 | **1.207**\*\*\* | 0.071 | **1.2**\*\*\* | 0.071 |
| Machiavellianism | | -0.36 | 0.281 | -0.272 | 0.28 | -0.300 | 0.289 |
| Positive emotion | | **-0.664**\* | 0.395 | **-0.739**\* | 0.405 | **-0.733**\* | 0.401 |
| Negative emotion | | -0.164 | 0.386 | -0.118 | 0.383 | -0.132 | 0.379 |
| Positive emotion\* expected partners' destruction (= nb of destructors) | 0 | | | **-1.36**\*\* | 0.622 | **-1.227**\*\* | 0.614 |
| | 1 | | | **-1.088**\*\* | 0.502 | **-1.011**\*\* | 0.496 |
| | 2 | | | **-0.817**\*\* | 0.419 | **-0.794**\* | 0.414 |
| | 3 | | | -0.545 | 0.396 | -0.578 | 0.393 |
| | 4 | | | -0.274 | 0.443 | -0.361 | 0.441 |
| | 5 | | | -0.003 | 0.542 | -0.145 | 0.540 |
| | 6 | | | 0.269 | 0.671 | 0.072 | 0.668 |
| | 7 | | | 0.54 | 0.814 | 0.288 | 0.811 |
| | 8 | | | 0.811 | 0.967 | 0.505 | 0.962 |
| | 9 | | | 1.082 | 1.124 | 0.721 | 1.118 |
| | 10 | | | 1.354 | 1.285 | 0.938 | 1.278 |
| Negative emotion\* expected partners' destruction (= nb of destructors) | 0 | | | 0.374 | 0.581 | 0.359 | 0.58 |
| | 1 | | | 0.159 | 0.469 | 0.144 | 0.467 |
| | 2 | | | -0.056 | 0.394 | -0.071 | 0.390 |
| | 3 | | | -0.272 | 0.378 | -0.286 | 0.374 |
| | 4 | | | -0.487 | 0.428 | -0.501 | 0.425 |
| | 5 | | | -0.702 | 0.525 | -0.716 | 0.524 |
| | 6 | | | -0.917 | 0.649 | -0.931 | 0.649 |
| | 7 | | | -1.132 | 0.786 | -1.146 | 0.788 |
| | 8 | | | -1.348 | 0.932 | -1.361 | 0.935 |
| | 9 | | | -1.563 | 1.082 | -1.576 | 1.086 |
| | 10 | | | -1.778 | 1.236 | -1.790 | 1.241 |
| *Control variables* | | | | | | | |
| Risk Averse | | | | | | -0.442 | 0.363 |
| Ambiguity Averse | | | | | | -0.232 | 0.332 |
| Prosocial | | | | | | 0.009 | 0.318 |
| Gender | | | | | | **-0.591**\* | 0.334 |
| Experience | | | | | | **1.067**\*\* | 0.527 |
| **Log Likelihood** | | -254.02 | | -249.54 | | -245.22 | |
| **LR Chi2** | | **196.31**\*\*\* | | **205.28**\*\*\* | | **213.91**\*\*\* | |
| **Pseudo R2** | | 0.279 | | 0.291 | | 0.304 | |
| **Number of observations** | | 172 | | 172 | | 172 | |

\*p < .1

\*\*p<0.05

\*\*\*p<0.01; SD: standard deviation; AME: average marginal effect

A second result reports that Machiavellianism predicts decisions made in JoD. The result is consistent with a recent work by Marcus et al. [65], who found that spitefulness is associated with high scores on Machiavellianism. It is to note that Machiavellianism is typically defined by high levels of manipulation, self-interest and spitefulness.

A third finding shows that negative incidental emotions seem not to play any direct effect on the willingness to destroy partner's endowment, whereas positive incidental emotions increase destructive behavior for individuals with high Machiavellian traits.

Contrary to negative emotions that narrow the scope of attention and make people focus on details, positive emotions broaden the scope of attention and make people engage in superficial analyses of information [100, 101]. Similarly, Gasper & Clore [102] suggest that positive emotions enhance *a greater focus on the forest* while negative emotions promote *a greater focus on the trees*. We argue that individuals will use the experienced emotion (positive emotions) as a source of information about the decision at hand. In fact, they globally evaluate the situation as favorable and not threatening, and therefore might have decided not to destroy others' money. Not surprisingly, these arguments do not hold for individuals with high Machiavellian traits as affective coldness is one of the main features of Machiavellianism. Indeed, they display a lack of emotional sensitivity and show deficits in understanding emotions [103].

It is to mention that our sample is only able to detect an effect size of 0.4. So, the fact that we cannot statistically confirm the tendency that a negative mood reduces the incentive to destroy partner's endowment, is either because negative emotions don't influence destruction at all or because the impact is really small.

Another interesting finding underlines an indirect effect of positive emotions on destruction that passes through 2nd order beliefs. Indeed, positive emotions act as a mediator on 2nd order beliefs. As 2nd order beliefs influence positively 1st order beliefs, and 1st order beliefs influence positively willingness to destroy, we deduce that individuals in whom positive emotions were induced destroy less than others. In other words, our results suggest that positive emotions promote positive expectations about others' behaviors and beliefs. We argue that happy individuals, when judging others' beliefs, might be more prone to naive realism, and since they are optimists in their expectations about others and believe that others share these beliefs, they choose to not destroy. Indeed, as suggested by social psychology, people perceive the world through a lens of naive realism. Naive realism is a bias that has been documented for second-order beliefs whereby individuals tend to believe that their own beliefs and judgments are more common and suitable than other responses. They believe that they do that in an objective and unbiased way [104]. They consequently often overestimate the degree to which their beliefs are shared by many others [105]. In addition, based on research suggesting a link between incidental emotions and the assessment of unknown probabilities of potential events, happy individuals make more optimistic probabilistic judgments while sad individuals made more pessimistic judgments [106].

In our experiment we considered emotions to vary according valence, making people either feeling sad (negative valence) or sad (positive valence). Nonetheless, there is a range of specific emotions in either category which might be more incline to elicit antisocial or prosocial behavior, for example by focusing on specific emotions such as anger or empathy. Moreover, extending the experimental game to both positive (pro-social) and negative (anti-social) behavior would give a larger view of the effect of emotions.

In a similar vein, assertions about the effects of emotions on decision making were found to be contingent on personality traits. In truth, these processes rarely apply to all personalities equally. An interesting extension would be to test whether certain populations are more prone to destruction, and whether their expectations differ also according to the type of partner. For example, will participants adopt similar behavior, and have similar expectations when their partner is either a man or a woman; a younger person or an older person; from the same religion or another religion; etc.

Extending the analysis to more sophisticated destruction games such as Fragile Public Good Game [107] will allow to study the effect of emotions both on constructive and destructive behavior, and see how behavior evolves over time.

## Supporting information

**S1 File.**
(DOCX)

**S2 File.**
(DOCX)

**S1 Data.**
(DTA)

## Author Contributions

**Conceptualization:** Lisette Ibanez, Hayet Saadaoui.

**Data curation:** Lisette Ibanez, Hayet Saadaoui.

**Funding acquisition:** Lisette Ibanez.

**Investigation:** Hayet Saadaoui.

**Methodology:** Lisette Ibanez, Hayet Saadaoui.

**Writing – original draft:** Hayet Saadaoui.

**Writing – review & editing:** Lisette Ibanez.

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
