## [Decision Letter · Decision Letter 0]

28 Jan 2022

PONE-D-21-35528An experimental investigation on the dark side of emotions and its aftereffectsPLOS ONE

Dear Dr. Saadaoui,

Thank you for submitting your manuscript to PLOS ONE. After careful consideration, we feel that it has merit but does not fully meet PLOS ONE’s publication criteria as it currently stands. Therefore, we invite you to submit a revised version of the manuscript that addresses the points raised during the review process.

We look forward to receiving your revised manuscript.

Kind regards,

Nikolaos Georgantzis, Dr.

Academic Editor

PLOS ONE

Journal Requirements:

2. Please include your full ethics statement in the ‘Methods’ section of your manuscript file. In your statement, please include the full name of the IRB or ethics committee who approved or waived your study, as well as whether or not you obtained informed written or verbal consent. If consent was waived for your study, please include this information in your statement as well

Reviewers' comments:

Reviewer's Responses to Questions

**Comments to the Author**

1. Is the manuscript technically sound, and do the data support the conclusions?

Reviewer #1: Partly

Reviewer #2: Partly

2. Has the statistical analysis been performed appropriately and rigorously? 

Reviewer #1: No

Reviewer #2: N/A

3. Have the authors made all data underlying the findings in their manuscript fully available?

Reviewer #1: Yes

Reviewer #2: Yes

4. Is the manuscript presented in an intelligible fashion and written in standard English?

Reviewer #1: Yes

Reviewer #2: Yes

5. Review Comments to the Author

Reviewer #1: Please find the attached file.

Reviewer #2: In this paper, the authors carry out a laboratory experiment in order to shed light on the effect of incidental emotions on virtuous behavior. Their main results are: (1) Machiavellian trais and expected destruction by partners drive destructive behavior; (2) emotional states influence destructive behavior indirectly.

Below, I list my major concerns follow by minor issues.

Major concerns:

1.In the experimental design section, authors explain that instructions were neutral and they change the term “destroy” by “reduce”. However, in the “Elicitation of first order and second order beliefs” they quote the sentences of the experiment and the word “destroy” appears. Additionally, these questions do not appear in the experimental instructions in that way.

2.It is necessary to introduce a sections name, for instance, materials and methods, where all the tasks of the experiment are explained in detail. If not, the readers that are not specialists can easily lose the thread. It will be very useful to explain all the controls introduced in the regressions.

3.The way of presenting the non-parametric test in table is quite confusing. Furthermore, no normality tests are presented do it is not justified the use of non-parametric tests.

-For instance, in table 2 authors have to specify that the Kruskall-Wallis test values reported are the statistics.

-In table 3, again it is necessary to specify that the values of the Mann-Whitney tests are the statistics. Additionally, a mistake is made before result 1. Authors write “Using a Mann-Whitney test, the hypothesis that the proportion of (…) cannot be rejected at p<0.05”. If the hypothesis cannot be rejected is because the p-value is higher than a 5% level of significance, not lower as it is stated.

-In the “Influence of emotions on players’ first and second-order beliefs”, authors run a Mann-Whitney test and conclude “positive and neutral conditions are not drawn from the same distribution”. This test is not a distributional test like for instance the Kolgomorov Smirnov, so this conclusions is not right.

-In table 4, I think that the specification of the Mann-Whitney test values is necessary. Also, it is striking to me that in table 4, authors find statistically significant differences (although one is at 10% level) between positive and neutral emotions and not between positive and negative and, on the other hand, between neutral and negative. What can be the explanation? Are the results affected by sample size? It will be a good option to run an ex-post power analysis to know if results are influenced by sample size.

-In the Expectations and Destructions, I think that a correlation matrix with significance levels is necessary to support the result.

-Regarding the regression analysis, I imagine that the coefficients of the model are not marginal effects. Thus, I think that authors must report marginal effects in these regressions, if not interpretation is not possible. In addition, the presentation of the table is not easy to read given that standard deviations are in the same line that coefficients. First, authors should specify that values between brackets are standard deviations in the same way that it is done for statistical levels.

-The results through which authors convey the argument of the paper is that positive emotions act as a mediator on second order beliefs. But, this variable is only statistically significant at a 10% level. A good way to explore the role of emotions and first and second order beliefs jointly is to include interactions in the logit model. In this way, authors will know if beliefs of those with positive emotions affect destruction as they conclude and it is clearer. Authors introduced interactions in the second and third models, but not in the first one.

-Authors include variable interactions in model 2 and 3 that are not properly explained.

Minor comments:

-The sections of the paper are not numbered, and it is not easy to know where they are included. The font size is similar, so sections and subsections are difficult to distinguish.

-There are some typos in the manuscript, so it must be revised. Some are grammatical mistakes, there is one sentence repeated in pages 18 and 19 and in the conclusion section (second paragraph, second line) authors must clear up one sentence.

-In figure 2 and 3, please assign labels to the axis.

6. PLOS authors have the option to publish the peer review history of their article (what does this mean?). If published, this will include your full peer review and any attached files.

Reviewer #1: No

Reviewer #2: No

---

## [Author Response · Author response to Decision Letter 0]

22 Apr 2022

Review for article PONE D 21 35528

An experimental investigation on the dark side of emotions and its aftereffects

Dear Editors, 

Thank you for giving us the opportunity to submit a revised draft of our manuscript titled “An experimental investigation on the dark side of emotions and its aftereffects” to PLOS ONE. We appreciate the time and effort that you and the reviewers have dedicated to providing your valuable feedback on our manuscript. We are grateful to the reviewers for their insightful comments on our paper. We have been able to incorporate changes to reflect most of the suggestions provided by the reviewers. We have highlighted the changes within the manuscript. 

Here is a point-by-point response to the reviewers’ comments and concerns

 

Reviewer #1: 

Summary:

This paper aims at investigating the role of beliefs and incidental emotions on destruction behavior in a Joy of Destruction experimental game. The two main determinants of destruction are spite and preemptive retaliation (negative reciprocity based on beliefs). It is found that positive incidental emotions have no direct impact on behavior and on beliefs on partners’ behavior (1 st order beliefs). However, they have a direct impact on beliefs on partners’ beliefs (2 nd order beliefs), which in turn influences positively 1 st order beliefs and eventually behavior. Hence positive incidental emotions indirectly slightly reduce destruction behavior.

Overall assessment

This is an interesting and well-written article. The research question is relevant and original because it deals with (1) bad behavior instead of virtuous behavior, (2) incidental instead of experienced emotions, (3) 2 nd order beliefs and not only 1 st order beliefs.

I have however important concerns that I would like to be handled by the authors before recommending publication:

- First, the experimental procedures appear to be correct, although the design should be better justified. Overall, the links between the theoretical section, the experimental design section and the results section should be clearer.

- Second, I am not fully convinced by the statistical analysis for now, especially the econometric part. The models and methods used should be better described and justified. Explanatory variables are not correctly defined. The interpretations of some results are debatable. At the end of the day, it seems to me that the conclusion of the article overstates the results of the experiment: the impact of incidental emotions is perhaps not as strong as seemed to be claimed.

We sincerely thank you for your time, and moreover, your useful remarks and constructive questions that helped us to improve this revised version of the manuscript. We have substantially modified and rewritten the paper.

Please find below answers (in blue) to your raised concerns (in black).

Detailed comments:

Introduction

- Overall, it is well written.

- Typos:

Page 6: Strange phrase: “We show that choosing to destroy others’ endowment is, at least to some, rational that is based on expectations.”

We corrected this sentence.

Theoretical framework and behavioral hypotheses

- Page 9: Could Hypothesis 1 be made more precise with respect to the model, stating clearly which variables of the model are involved? In other words, does Hypothesis 1 mean that the probability 𝑗 of destruction increases with 𝑃 𝑖 and decreases with 𝑆 𝑖 and 𝛼 𝑖 ? Indeed, the term “degree of spitefulness” used in the hypothesis is a bit ambiguous.

Thank you for this suggestion. We have rewritten Hypothesis 1 and included the model’s parameters.

- Page 10: Hypothesis 3 is convincing and the justifications are fine. However, Fochmann et al. (that the authors state just before) seem to go in the opposite direction: “Similarly, Fochmann et al. [27] provide evidence that individuals have a lower willingness to comply with social norms after being primed with positive incidental emotions compared with aversive emotions.” Arguably, following this argument, one might expect that after positive incidental emotion, subjects might be willing to destroy more? In other words, Hypothesis 3 seems to be intuitive but are there counterarguments in the literature?

We agree with you that results related to the Fochmann et al. reference are ambiguous. In fact the authors focus on tax compliance and show that participants under positive emotions fraud more. One of the interpretations of this result is “avoiding norm compliance”. We argue that this result might have other explanations, notably that emotions shape expectations. For instance, being positively mooded, may reduce expectations to be audited. 

(see page 9)

- Is it possible to make hypotheses regarding which variables of the model are impacted by incidental emotions? Will the impact be on 𝑃 𝑖 ? 𝑆 𝑖 ? Or on the 1 st order belief? Or on the 2 nd order belief? That is, are people directly affected by incidental emotions or do they believe that others will be? Maybe it would be relevant to introduce the impact of emotions into the mathematical model?

We followed your suggestion to me more precise concerning the impact of incidental emotions on the motivators of destructive behavior, i.e. spitefulness and preemptive retaliation (new Hypothesis 2). We didn’t introduce the emotional state into the mathematical model (because it is considered to be an exogenous shock), but indicated the impact of emotions on the parameters in the hypotheses. 

In hypothesis 3 (former hypothesis 2), we discuss the relevance of expectations on destructive behavior, and more precisely the role of 2nd order beliefs. However, we are not able to predict how these expectations evolve with different emotional states.

- Typos:

Page 8, top: Maybe this terminology is standard in this literature but to me, 𝑆 𝑖 should not be referred to as “spitefulness”, this is reciprocity (more specifically retaliation). 𝑃 𝑖 could be referred to as spitefulness.

Thank you very much for pointing these issues, and we absolutely agree with your suggestion. Indeed, spitefulness is the opposite of altruism (cf. Levine, 1998) and therefore doesn’t depend on other’s (expected) behavior. We renamed the variables.

Ref: Levine DK (1998) Modeling altruism and spitefulness in experiments. Review of Economic Dynamics, 1(3), 593-622.

Experimental design

- The experimental procedures seem to be very good. The protocol allows to collect a variety of data. Nevertheless, this section is very descriptive overall. The authors should not only describe but also briefly justify their design: for example, why elicitating risk preferences? Social value orientation?... What are the conjectures regarding these variables? For example, is a more riskaverse subject expected to destroy more? (this is what I would say intuitively: in the absence of destruction, the variance of gains is large, whereas destructing is like a costly insurance as it provides a “revenge utility” which partly compensates the potential harm of the other player) And why using these specific procedures instead of others? (e.g., why this risk aversion test and not Holt-Laury?) For sure, the text should however be kept concise.

We followed your suggestion to add concise motivations for the elicitation of our control variables. 

We have incorporated the following paragraph in the manuscript (page 13).

“Because social preferences are of crucial importance in understanding interdependent decision-making behavior (Böhm et al., 2017), we wish to explore their influence in our experiment (Task 3). For this reason, we elicit subjects’ social value using the social value orientation (SVO) slider measure developed by Murphy et al. (2011). Social preferences based on the SVO framework have been found to be predictive to behavior in many games (e.g., Balliet et al., 2014). For instance, some studies show a positive association between individuals’ general level of pro-sociality and their willingness to engage in inter-group conflict (Aaldering et al., 2013). 

In Task 4, we elicited risk and ambiguity aversion using Ellsberg urns (Ellsberg, 1961) since several forms of antisocial behavior are found to be linked to decisions made under conditions of ambiguity and risk (Buckholtz et al., 2017). The choice of this specific procedure is justified by its ability to test both ambiguity and risk at the same time contrarily to other procedures. “

At this stage we didn’t include any conjectures but discussed the results in line with findings in the literature in section 5.

- Typos:

Page 12: I guess that the social value orientation game is Part 2 in the instructions? If so, the sequence presented in Table 1 does not seem to match exactly with the sequence presented in the instructions.

Thank you very much for pointing this issue. We modified Table 1 to describe exactly the sequence presented in the instructions. 

Results

- Table 2: aren’t the authors also interested in the elicited level of arousal (as well as valence)?

You raise an interesting issue. We modified Table 3 (former Table 2) considering both valence and arousal, and adapted the text.

- The treatment comparison tables are not very easy to read. For example, in Table 3, what is exactly the “-0.467”? It is possible to guess after some thought, but authors might be willing to help the reader in notes below the tables. Figures 2 and 3 are also not very clear (what are the red curves?). In Table 4, I do not understand exactly what the “4.430” and “3.613” are; I would expect them to be negative (?).

We agree that the way results were presented might be confusing. Consequently, we have reorganized the tables to be easier to read.

Table 4 describes treatment-specific summary statistics.

And Table 5 describes z-statistics of pairwise Mann–Whitney statistical tests to analyze for differences in destructive behavior and expectations between emotional states. 

And indeed, the values “4.430” and “3.613”, they are negative.

In Figure 2: “Expectations of others’ behaviors & destruction”, we show the relationship between Destruction and Expectations of others’ behaviors (First order belief) graphically. So, the Lowess smoother curve shows that a change in destruction from zero to one increases with an increase of one’s expectations of destruction by others. 

Accordingly, we incorporated the following sentence in the text: 

“To present the underlying nature of the relationship between destruction and expectations, Lowess smoother curves were used.”

- Page 16: I’m not convinced by the way Result 2 is written: “Positive incidental emotions lower individual’s expectations of other players’ destructive behavior and beliefs.” → You should mention that it is relative to the neutral emotion since no significant difference with the negative emotion treatment shows up. It is surprising that negative emotions have no significant impact, and if anything, that they tend to curb decisions and beliefs in the same direction as positive emotions. Would the authors have explanations for this?

We absolutely agree with your suggestion and changed Result 2 to: 

“Compared to neutral emotions, positive emotions lower individual’s expectations on partners’ destructive behavior and beliefs.”

We also included conjectures for negative emotions in Hypothesis 2. However, results seem to go in the opposite way.

An explanation that negative emotions (compared to neutral emotions) also decrease incentives to destroy and to expect others to be less destructive might be explained by the moral compensation phenomenon, or more precisely by moral cleansing. Moral cleansing describes behaviors aimed at restoring moral self-worth in response to past transgressions (West and Zhong, 2015). In our specific case the negative emotional induction might have reduced the self-worth, and therefore participants might have lower incentives to destroy than in the neutral emotion treatment, but also anticipate lower destruction by others.

See for instance: West, C., Zhong, C. 2015. ”Moral cleansing , Current Opinion in Psychology, 6: 221-225.

- Overall, the econometric analysis may be more detailed. For example, are you using specific econometric methods? What are the assumptions regarding the variance-covariance matrix of the error term? Why a Tobit regression? How many left- and right censored values are there? The explanatory variables are not defined (what is the definition of variable “gender” for example? What is “risk aversion”? What about the measure of ambiguity aversion?) and descriptive statistics on these variables should be provided. Table 5, model (2): it seems to me that getting a pseudoR² larger than 1 reveals that the model fits the data poorly. Could the authors check this?

We have performed a Skewness and Kurtosis test for normality and results indicate that residuals are not normally distributed (p<0.5). We therefore use non-parametric tests, and Logit and censored Tobit regressions. We observe 65, respectively 66 left censored values, and only 2, resp. 3 right censored values, for the 1st order beliefs, respectively 2nd order beliefs.

We included a table with description of all the explanatory variables (Table 2), and included an ambiguity aversion measure.

- Table 5, models (2) and (3): it could be clearer to present first regressions without interaction terms since they complicate interpretations. For example, in model (3), the term “Positive emotion” is significantly negative. But because of the presence of the term “Positive*Destruction”, it means that the former should be interpreted as the effect of positive emotions for the subjects who do not destroy. It seems actually that the global marginal effect of “Positive emotion” (0.075+0.0885) is actually very small?

We now introduce interaction terms, only after having interpreted a model without any interaction term.

- …Still Table 5: I don’t know if it is a good idea to include “Destruction” into models (2) and (3) since it looks like a circular reasoning: beliefs are supposed to be a determinant of destruction, not the opposite. Beware of endogeneity issues. (Same issue with “Expected partner’s Behavior” in model (3).)

Thank you for pointing this inconvenience We have followed your suggestion and, in order to avoid the circular reasoning, we now explain destruction according 1st order and 2nd order beliefs, 1st order beliefs according 2nd order beliefs, and 2nd order beliefs only according emotional states and participant-specific variables.

- …Still Table 5: some results are simply stated without much explanation although they appear to be counterintuitive. For example, “participants who inflict destruction to their partners expect their partners to believe they will adopt less destructive behavior than will expect participants who decide not to destroy the endowment of their partner (Model 3).” Other results are ignored, such as for example the effect of risk aversion. More risk-averse subjects expect their partners to believe they will adopt less destructive behavior. Is it possible to find some explanation for this?

We now discuss all results in more detail. Moreover, we present and discuss marginal effects.

- Structural equation model:

The estimation procedure used for the structural equation model should be more detailed. I am wondering whether model (4) would not be sufficient instead of models (1) to (3), since it would avoid circular reasoning as mentioned before.

Why are there so few explanatory variables in the second regression? It was shown in regression (3) that positive emotion has a negative impact on partner’s expectation. So, why not considering a three-equation structural model, regressing 2 nd order beliefs on various control variables such as emotion, then regressing 1 st order beliefs on the 2 nd order beliefs and on same set or other control variables, then regressing willingness to destroy 1 st order beliefs and on the same set of variables or others? The first regression should allow to show that emotions impact significantly 2 nd order beliefs.

As stated above, we choose to drop the structural equation model as it is not well adapted to our data structure. We followed your suggestion to be more rigorous in our choice of explanatory variables and avoid circular reasoning. With the new models we still capture the indirect effects on destruction that passes through first-order beliefs and second order beliefs.

- Typos:

Page 18-19: The sentence “Participants who have already participated in lab experiments have lower expectations on their partner’s willingness to destroy.” is written twice.

Thank you for pointing this mistake.

Discussion & conclusion

- The main result is interesting: incidental emotions would not impact destruction behavior directly, or through 1 st order beliefs, but only 2 nd order beliefs. Are there other studies on the impact of emotions on 2 nd order beliefs? Are there some psychological or neurological explanations or hypotheses regarding this phenomenon?

Thank you for pointing this issue. We argue that happy individuals, when judging others’ beliefs, might be more prone to naive realism, and since they are optimists in their expectations about others and believe that others share these beliefs, they choose to not destroy. Indeed, as suggested by social psychology, people perceive the world through a lens of naive realism. Naive realism is a bias that has been documented for second-order beliefs whereby individuals tend to believe that their own beliefs and judgments are more common and suitable than other responses. They believe that they do that in an objective and unbiased way (Ross and Ward, 1996). They consequently often overestimate the degree to which their beliefs are shared by many others (Egan et al. 2014). In addition, based on research suggesting a link between incidental emotions and the assessment of unknown probabilities of potential events, happy individuals make more optimistic probabilistic judgments while sad individuals made more pessimistic judgments (Wright & Bower, 1992). 

- The conclusions should be qualified given Result 1: there seems to be an effect of incidental emotion, but this effect appears to be small anyway. Overall, the conclusion seems to slightly overstate the results of the paper. For example, the phrase “Indeed, positive emotions induce people to be more optimistic on others’ behavior and beliefs and thus reduces preemptive retaliation” is correct based on the Mann-Whitney comparison tests but appears to be less clear from the econometric analysis (i.e., after controlling for various factors).

We agree that we slightly overstated our former results. With the new econometric analyses results seem to be more robust and show that positive emotions impact destruction both directly and indirectly. In the conclusion, we now present and discuss main results. 

- Typos:

The first sentence of the conclusion is strangely written.

We corrected. 

Page 22: “we deduced that individuals in whom positive emotions were induced TO destroy less than others.”

We corrected.

Instructions

- Typo: “Experiment” instead of “experience”.

We corrected as suggested. 

 

Reviewer #2: 

In this paper, the authors carry out a laboratory experiment in order to shed light on the effect of incidental emotions on virtuous behavior. Their main results are: (1) Machiavellian trais and expected destruction by partners drive destructive behavior; (2) emotional states influence destructive behavior indirectly.

Below, I list my major concerns follow by minor issues.

We sincerely thank you for your time, and moreover, your useful remarks and constructive questions that helped us to improve this revised version of the manuscript. We have substantially modified and rewritten the paper.

Please find below answers (in blue) to your raised concerns (in black).

Major concerns:

1.In the experimental design section, authors explain that instructions were neutral and they change the term “destroy” by “reduce”. However, in the “Elicitation of first order and second order beliefs” they quote the sentences of the experiment and the word “destroy” appears. Additionally, these questions do not appear in the experimental instructions in that way.

Thank you for pointing this issue. We apologize for this negligence. We corrected for both inconsistencies.

2.It is necessary to introduce a sections name, for instance, materials and methods, where all the tasks of the experiment are explained in detail. If not, the readers that are not specialists can easily lose the thread. It will be very useful to explain all the controls introduced in the regressions.

We corrected as suggested. 

3.The way of presenting the non-parametric test in table is quite confusing. Furthermore, no normality tests are presented do it is not justified the use of non-parametric tests.

Thank you for pointing this issue. We have performed a Skewness and Kurtosis test for normality and results indicate that residuals are not normally distributed (p<0.5). We now indicate this in the manuscript. 

We also modified the former Tables 2 and 3, now Tables 4 and 5.

-For instance, in table 2 authors have to specify that the Kruskall-Wallis test values reported are the statistics.

We corrected as suggested. 

Table 4 describes treatment-specific summary statistics, and Table 5 describes z-statistics of pairwise Mann–Whitney statistical tests to analyze for differences in destructive behavior and expectations between emotional states. 

-In table 3, again it is necessary to specify that the values of the Mann-Whitney tests are the statistics. 

We corrected as suggested. 

Additionally, a mistake is made before result 1. Authors write “Using a Mann-Whitney test, the hypothesis that the proportion of (…) cannot be rejected at p<0.05”. If the hypothesis cannot be rejected is because the p-value is higher than a 5% level of significance, not lower as it is stated.

Thank you for pointing this out. We corrected as suggested. 

-In the “Influence of emotions on players’ first and second-order beliefs”, authors run a Mann-Whitney test and conclude “positive and neutral conditions are not drawn from the same distribution”. This test is not a distributional test like for instance the Kolgomorov Smirnov, so this conclusions is not right.

Thank you for pointing this out. We agree that the statement is wrong and thus corrected as suggested. 

“A Mann-Whitney test (Table 5) gives further support that individuals’ expectations about other players’ behavior, in the positive and neutral conditions, are significantly different (z-statistic= -2.09, p<0.05).”

-In table 4, I think that the specification of the Mann-Whitney test values is necessary. Also, it is striking to me that in table 4, authors find statistically significant differences (although one is at 10% level) between positive and neutral emotions and not between positive and negative and, on the other hand, between neutral and negative. What can be the explanation? Are the results affected by sample size? It will be a good option to run an ex-post power analysis to know if results are influenced by sample size.

We agree and indicate that values are z-statistics. 

We run an ex-post power analysis using the G*power software to determine the minimum sample size required. The analysis revealed that a minimum sample size of 48 observations allows to obtain a power of .80 (with alpha=0.05).

An explanation that negative emotions (compared to neutral emotions) also decrease incentives to destroy and to expect others to be less destructive might be explained by the moral compensation phenomenon, or more precisely by moral cleansing. Moral cleansing describes behaviors aimed at restoring moral self-worth in response to past transgressions (West and Zhong, 2015). In our specific case the negative emotional induction might have reduced the self-worth, and therefore participants might have lower incentives to destroy than in the neutral emotion treatment, but also anticipate lower destruction by others.

See for instance: West, C., Zhong, C. 2015. ”Moral cleansing , Current Opinion in Psychology, 6: 221-225.

-In the Expectations and Destructions, I think that a correlation matrix with significance levels is necessary to support the result.

Thank you for pointing this issue. We performed several analyses. First, we checked for collinearity of 1st and 2nd order beliefs, and made sure that both could be maintained in the regression (variance inflation factor < 4). Second, we find a high correlation between Expected partners’ destruction and Expected partners’ beliefs on destruction (Spearman’s rho=0.87, p<0.1), which is confirmed in Models 3 and 4.

-Regarding the regression analysis, I imagine that the coefficients of the model are not marginal effects. Thus, I think that authors must report marginal effects in these regressions, if not interpretation is not possible. In addition, the presentation of the table is not easy to read given that standard deviations are in the same line that coefficients. First, authors should specify that values between brackets are standard deviations in the same way that it is done for statistical levels.

We absolutely agree and indicate marginal effects directly in the tables. We also reorganized the presentation of the tables, and now specify that the standard deviations are reported in brackets.

-The results through which authors convey the argument of the paper is that positive emotions act as a mediator on second order beliefs. But, this variable is only statistically significant at a 10% level. A good way to explore the role of emotions and first and second order beliefs jointly is to include interactions in the logit model. In this way, authors will know if beliefs of those with positive emotions affect destruction as they conclude and it is clearer. Authors introduced interactions in the second and third models, but not in the first one.

Thank you for your suggestion. We now only use Logit and censored Tobit regressions, as residuals are not normally distributed (Skewness and Kurtosis test for normality, p<0.5).

To be more rigorous in our choice of explanatory variables and avoid circular reasoning, we explain destruction according 1st order and 2nd order beliefs, 1st order beliefs according 2nd order beliefs, and 2nd order beliefs only according emotional states and participant-specific variables. With the new models we still capture the indirect effects of positive emotions on destruction that passes through first-order beliefs and second order beliefs, and is now statistically significant at a 5% level.

We also include an interaction term (Machiavellianism*positive emotion) in the logit regression (which explains the determinants of destructive behavior); and an interaction term (Positive emotion*expected partners’ belief) in the censored Tobit regression to explain expected partners’ destruction. These interaction terms allow to test the (indirect) impact of emotions on parameters of the theoretical model. 

-Authors include variable interactions in model 2 and 3 that are not properly explained.

We now explain in more detail the interaction terms, especially in relation to the interpretation of results.

Minor comments:

-The sections of the paper are not numbered, and it is not easy to know where they are included. The font size is similar, so sections and subsections are difficult to distinguish.

Thank you for pointing this issue. We introduced numbers into the titles of sections.

-There are some typos in the manuscript, so it must be revised. Some are grammatical mistakes, there is one sentence repeated in pages 18 and 19 and in the conclusion section (second paragraph, second line) authors must clear up one sentence.

We corrected as suggested. 

-In figure 2 and 3, please assign labels to the axis.

We corrected as suggested.

---

## [Decision Letter · Decision Letter 1]

13 May 2022

PONE-D-21-35528R1An experimental investigation on the dark side of emotions and its aftereffectsPLOS ONE

Dear Dr. Saadaoui,

Thank you for submitting your manuscript to PLOS ONE. After careful consideration, we feel that it has merit but does not fully meet PLOS ONE’s publication criteria as it currently stands. Therefore, we invite you to submit a revised version of the manuscript that addresses the points raised during the review process, especially the detailed suggestions by the second reviewer.

We look forward to receiving your revised manuscript.

Kind regards,

Nikolaos Georgantzis

Academic Editor

PLOS ONE

Reviewers' comments:

Reviewer's Responses to Questions

**Comments to the Author**

1. If the authors have adequately addressed your comments raised in a previous round of review and you feel that this manuscript is now acceptable for publication, you may indicate that here to bypass the “Comments to the Author” section, enter your conflict of interest statement in the “Confidential to Editor” section, and submit your "Accept" recommendation.

Reviewer #1: (No Response)

Reviewer #2: (No Response)

2. Is the manuscript technically sound, and do the data support the conclusions?

Reviewer #1: Partly

Reviewer #2: Partly

3. Has the statistical analysis been performed appropriately and rigorously? 

Reviewer #1: Yes

Reviewer #2: Yes

4. Have the authors made all data underlying the findings in their manuscript fully available?

Reviewer #1: No

Reviewer #2: Yes

5. Is the manuscript presented in an intelligible fashion and written in standard English?

Reviewer #1: Yes

Reviewer #2: Yes

6. Review Comments to the Author

Reviewer #1: Please see the attached file.

Reviewer #2: 1.Authors have corrected the inconsistency between “reduce” and “destroy” in the text, but I am not completely sure about the term used in the experiment given that the questions that appeared in the original submissions seemed to be a copy paste from the experiment. Do you authors screen shots of the experiment?

2.Authors use dummies to control for risk and ambiguity aversion. Explain what is considered by authors to be a risk averse and ambiguity averse subject.

3.In table 4, authors state that “destruction is lower in the positive emotion treatment as compared to the negative emotion treatment and the neutral emotion treatment”, but the p-value is 0.78. Then, there are not statistically significant differences between these three treatments. In this table, the differences appear regarding the expectations on partners’ destruction although at a 10% level.

4.After Table 5, authors state that “the hypothesis that the proportion of individuals who destroy another player’s money is significantly different under the “positive”, “negative” and “neutral” emotional state is rejected at the 10% level”. The specification of this sentence seems to explain a Kruskal-Wallis test. In this case, there are binary combinations in which the null hypothesis points out towards no differences between the treatments compared. I do not know in which comparison a 10% level appears. In any case, the null hypothesis is always equality of medians, not differences. The result’s explanation is right, but not the previous argument.

5.Authors state that “Even if we cannot statistically confirm the tendency that a positive mood reduces the incentive to destroy partner’s endowment, we also observe this tendency for negative emotions contrarily to our hypothesis 2. An explanation that negative emotions (compared to neutral emotions) also decrease incentives to destroy might be explained by the moral compensation phenomenon, or more precisely by moral cleansing” With the data of Table 5 we cannot argue that there are differences between negative and neutral emotions in any of the variables compared.

6.Regarding the ex-post power analysis, what are the value authors have used to calculate it? You present it at the beginning, but the ex-post power analysis needs to be performed for all the test without statistically significant differences. That is, in Table 4 for all the comparison except for the ones with statistically significant results. Moreover, at the beginning you indicate that “a sample size of 50 observations allow us to obtain sufficient power”. I imagine that is the analysis is properly performed you will need 50 observations in each group.

7.After result 2, the sentence beginning by “Again” needs to me restructured. It difficult to understand the argument that authors try to convey. I do not see differences regarding negative emotions.

8.In table 6b, coefficients are not necessary if you provide marginal effects. Then, you can substitute them and at the end on the table you must specify that values are marginal effects, standard deviations between brackets and the levels of statistically significance. Please, remove coefficients and provide marginal effects for all the variables.

9.Given that tables 6a and 6b are measuring the same all the models must be presented in one single table. The first model will have the three explanatory variables included in the reviewed version. The second model will include positive and negative emotion. The third one will include the interactions and the third one will add the control variables. In this case, the models will be adding new variable to see the stability of results. Authors have not controlled for the emotions in the first model, have deleted one main explanatory variable in model 3 and have not controlled for individual characteristics in models 3 and 4. Please, provide the raw data and STATA commands.

10.At the expense of the changes suggested to the models, the variable “positive emotion” does not refer only to those subjects with low Machiavellian scores. This variable refers to all the subjects in the sample, then, a positive emotions reduced the probability of engaging in destructive behavior. But positive emotions increase destruction of those scoring higher in Machiavellian. Be careful with interpretations and with the sentences about the results.

11.Similarly to my suggestion about the models of Table 6, please run the same models adding variables step by step. Again, provide marginal effects. Please, rewrite the interpretation of positive emotions and the interaction after result 5, it is not clear.

12.Why do not the authors include the variable about first order beliefs in Table 8?

13.In order to check directly result 6, authors can include in Table 6 an interaction between second order beliefs and emotions.

14.After Table 8, there is an explanation about table 7. Please, move this explanation after Table 7. I really think that is not accurate to use the term “optimistic”. This result just points out that the fact of having participated in lab experiments previously reduces the expectation of others reducing their partner’s income.

15.Regarding the explanation about the gender effect, do not use the term “optimistic” and explain it again.

7. PLOS authors have the option to publish the peer review history of their article (what does this mean?). If published, this will include your full peer review and any attached files.

Reviewer #1: No

Reviewer #2: No

---

## [Author Response · Author response to Decision Letter 1]

5 Aug 2022

Review for article PONE ID 21 35528

An experimental investigation on the dark side of emotions and its aftereffects

Dear Editors, 

Thank you for giving us the opportunity to submit a revised draft of our manuscript titled “An experimental investigation on the dark side of emotions and its aftereffects” to PLOS ONE. We appreciate the time and effort that you and the reviewers have dedicated to providing your valuable feedback on our manuscript. We are grateful to the reviewers for their insightful comments on our paper. We have been able to incorporate changes to reflect most of the suggestions provided by the reviewers. We have highlighted the changes within the manuscript. 

Here is a point-by-point response to the reviewers’ comments and concerns

Reviewer 1

An experimental investigation on the dark side of emotions and its aftereffects

Summary:

This paper aims at investigating the role of beliefs and incidental emotions on destruction behavior in a Joy of Destruction experimental game. The two main determinants of destruction are spite and pre-emptive retaliation (negative reciprocity based on beliefs). It is found that positive incidental emotions have no direct impact on behavior and on beliefs on partners’ behavior (1st order beliefs). However, they have a direct impact on beliefs on partners’ beliefs (2nd order beliefs), which in turn influences positively 1st order beliefs and eventually behavior. Hence positive incidental emotions indirectly reduce destruction

behavior.

Overall assessment

I think that the authors did great job in improving the paper. I have still a lot of comments, most of which can now be considered as minor, except maybe the two comments that are presented first below, regarding subtle effects of positive incidental emotions on behavior and beliefs: I think you need to 1) better estimate them, 2) propose an interpretation for them.

We would like to thank you again for your careful reading, interesting suggestions and support in helping us to improve our paper. For convenience, we reproduce each of your comments below, followed by our responses in blue.

Detailed comments:

Main issues

- Page 22, “Result 4: Positive emotions reduce destruction for individuals with low traits of

Machiavellianism.” → I think the support for this result should be made stronger. You derive it from the results of the regression done in model 4 (table 6b). You state: “Indeed, the variable Positive emotions as well as the interaction term between Positive emotions and Machiavellianism being significant, means that for individuals with low Machiavellian scores, positive emotions lower their willingness to destroy partner’s endowment, whereas for people with high scores destruction increases.” However, I think you should compute proper marginal effects to reach these conclusions. For example, compute

𝑃𝑟𝑜𝑏(𝐷𝑒𝑠𝑡𝑟𝑢𝑐𝑡𝑖𝑜𝑛 = 1| 𝑃𝑜𝑠𝑖𝑡𝑖𝑣𝑒 𝑒𝑚𝑜𝑡𝑖𝑜𝑛𝑠 = 1, 𝑀𝑎𝑐ℎ𝑖𝑎𝑣𝑒𝑙𝑙𝑖𝑠𝑚 = 1) −

𝑃𝑟𝑜𝑏(𝐷𝑒𝑠𝑡𝑟𝑢𝑐𝑡𝑖𝑜𝑛 = 1| 𝑃𝑜𝑠𝑖𝑡𝑖𝑣𝑒 𝑒𝑚𝑜𝑡𝑖𝑜𝑛𝑠 = 0, 𝑀𝑎𝑐ℎ𝑖𝑎𝑣𝑒𝑙𝑙𝑖𝑠𝑚 = 1)

to properly estimate the effect of positive emotions for low Machiavellian subjects. You will

probably confirm Result 4. I would be curious to know the effect for highly Machiavellian subjects

𝑃𝑟𝑜𝑏(𝐷𝑒𝑠𝑡𝑟𝑢𝑐𝑡𝑖𝑜𝑛 = 1|𝑃𝑜𝑠𝑖𝑡𝑖𝑣𝑒 𝑒𝑚𝑜𝑡𝑖𝑜𝑛𝑠 = 1, 𝑀𝑎𝑐ℎ𝑖𝑎𝑣𝑒𝑙𝑙𝑖𝑠𝑚 = 5) −

𝑃𝑟𝑜𝑏(𝐷𝑒𝑠𝑡𝑟𝑢𝑐𝑡𝑖𝑜𝑛 = 1|𝑃𝑜𝑠𝑖𝑡𝑖𝑣𝑒 𝑒𝑚𝑜𝑡𝑖𝑜𝑛𝑠 = 0, 𝑀𝑎𝑐ℎ𝑖𝑎𝑣𝑒𝑙𝑙𝑖𝑠𝑚 = 5), 

which is maybe significantly positive, meaning that highly Machiavellian subjects tend to destruct more under positive emotions. If so, why? It is not intuitive that positive emotions should lead to more destruction for this type of subjects.

Thank you for pointing this issue. We tried to give better support for our results. 

First, we now only indicate marginal effects. Second, the Machiavellian variable is a continuous variable (from 1.33 to 5), so the interpretation of the crossed variable (Machiavellian*Positive emotion) is that the higher the Machiavellian traits the more positive emotions have a significant impact on destructive behavior. 

We also tried to better expain the findings. Indeed, it is not to question the beneficial consequences that positive emotions, positive thinking and positive attitudes derive such as, cooperativeness, flexibility, altruism and life satisfaction, etc. (e.g., Schwarz, 2000; Forgas, 2000). Rather, numerous empirical studies have revealed the darker side of happiness. In this vein, Gruber et al. (2011) indicated that experiencing happiness is not always positive. People who are in the pursuit of happiness tend to be to be more depressed, miserable, and unhappy. According to Tan and Forgas (2010) there exists a positive and significant relationship between happiness and selfishness. Further, Tamir and Bigman (2014) demonstrate that positive emotions can have a negative impact on people, and that negative emotions can have a positive impact. Thus, it is assumed that the possibility for positive emotion to have a significant impact on increasing Machiavellian subjects’ destructive behavior cannot be eliminated. Adding to this, previous research has showed that individuals with high levels of Machiavellianism usually endorse ego-centric or antisocial notions of well-being where they tend to prioritize their personal happiness over other people’s happiness (Kajonius et al., 2015). Consequently, it is presumed that these persons have decided to do whatever it takes to keep the positive emotional state even if their actions will have negative consequences or/and violate their own moral values.

- Page 23: I have similar comments as above. On the basis of the regression of model 6, you state:

“Regarding the impact of emotion, we observe that positive emotions moderate expectations on destruction according expectations on partners’ belief. For individuals with low expectations on partners’ beliefs, positive emotions lower their expectations on partners’ destructive behavior, whereas for individuals with high expectations on partners’ beliefs, expectations that partners decide to destroy increases.” → Again, you need to prove this by computing proper marginal effects. Again, would you have an interpretation? It is not intuitive that positive emotions should lead expected partner’s destruction to increase.

Thank you for pointing this issue. We now perfom the econometric analyses properly as expectations for detruction and beliefs are integer variables, and again only report marginal effects. 

As explained in the previous point, positive emotions can have a negative impact on individuals’ thoughts and behaviors. It is worthy to mention that a growing body of research investigating the relationship between emotional experience and social information processing indicates that when people are performing a social judgment task the type of strategy they use may be affected by their momentary emotional state (for a review, see for example Bodenhausen, 1993). In the same vein, certain positive emotions lead people to rely more on highly accessible cognitions, such as beliefs, expectations, and stereotypes (e.g., Forgas & Fiedler, 1996). Importantly, happiness has been usually associated with the use of more superficial or cursory styles of thinking. For instance, some studies on mood and persuasion (e.g., Schwarz, Bless, & Bohner, 1991) support this evidence and document that prior to the presentation of a persuasive message, happy people are less affected by variations in argument quality. Indeed, happy people often prefer to base their reactions more on simple cues ( Petty, Gleicher, & Baker, 1991). Thus, we assume that, under a positive emotional state, individuals has adopted a superficial thinking that disabled them from engaging in close scrutiny of the situation. It appears that these individuals expected their partner’s destruction to increase because they has interpreted the suspicious situation as a threat to be handeled with precaution.

Minor issues

Introduction

- Page 2, second parag.: no comma after “While”. 

Corrected as suggested

Theoretical framework and behavioral hypotheses

- Page 6: “The reciprocity utility has two components: 𝑆𝑖 which correspond to individual i's

satisfaction of destruction or spitefulness and 𝑅𝑖𝛼𝑖𝑗 the reciprocation term which...” → No, the reciprocity utility does not include 𝑆𝑖.

Thank you for this remark, and we agree that S isn’t included in the reciprocity utility. We replaced “the reciprocity utility” by “the non-monetary utility of destruction”.

- Page 10: “Hypothesis 3: Participants’ second-order beliefs influence destruction through first-order beliefs.” → You could even state that the relation is expected to be increasing, right?

Thank you for this interesting suggestion, we modified Hypothesis 3 accordingly.

Experimental design

- Page 10: Regarding the power analysis. This is a good point but it seems to me that your statement is too imprecise. Power analysis requires that you specify the effect size that you are expecting.

Thank you for pointing this issue. We now indicate the expected effect size for our sample. We added the following sentences to the manuscript.

We ran an ex-post power analysis using the G*power software which revealed that our sample size (respectively 55, 57 and 60 observations for the three treatments) is sufficient (power=0.8) to observe an effect size of 0.4. While Cohen (1992) recommends that wherein d = 0.20 indicates a small effect, d = 0.50 indicates a medium effect and d = 0.80 indicates a large effect, Dancey & Reidy (2006) specify that small and medium effects are more willingly available when evaluating behavioral and psychological constructs as it is the case of constructs investigated herein. 

- Page 12: "We suppose second order beliefs to be a proxy for 𝑅, i.e., player’s willingness to reciprocate." → This sentence is misleading, why not: "We suppose second order beliefs to be a proxy for 𝑅, i.e., the other player’s willingness to reciprocate." However, even though I see what you mean, I am a bit confused with this interpretation which seems to come from nowhere. Why not presenting and discussing it properly in the theoretical section (if it is worth it)? In the model, beliefs (the alpha’s) and the willingness to reciprocate (the R’s) are separate parameters so this assumption is not so obvious.

Thank for very much for this suggestion. We now present R to be partner’s expectations on the other player’s willingness to reciprocate directly in the theoretical section. 

- Page 14, table 2: Regarding the variables such as Expected partner’s behavior, wouldn't they be more informative in relative terms? Besides, how many subjects are there in a session (I guess 10 from the following, but it seems that you never mention it explicitly). In addition, check the definition of Expected partner’s belief in the table. It seems to be wrong.

We have corrected as suggested.

Results

- Page 15:

o “We start by examining...” → This paragraph is not clearly structured. You should explain

that it is more relevant here to run diff. in diff. tests than simple diff. tests. This is not at first sight.

We agree and now discuss both the simple difference as well as the diff in diff tests.

o In your answers to my previous comments, you seemed to state that you used Logit and Tobit regressions to make up for non-normal error terms. However, these are still parametric analyses (thus, based on normality), so that I don’t think this is the correct justification. You are using Logit regressions because the dependent variable is binary and a Tobit regression because the dependent variable is left/right censored (specify the number of censored observations).

Thank you for pointing this out and we absolutely agree with your arguments. We now justify the censored Tobit regression indicating the number of censored observations.

- Table 3: Please specify the number of observations.

Corrected as suggested 

- Page 17: “Then, the hypothesis that the proportion of individuals who destroy another player’s money is significantly different under the “positive”, “negative” and “neutral” emotional state is rejected at the 10% level.” → The sentence seems to be wrongly stated, better: "The hypothesis of no difference across treatments cannot be rejected at 10%."

We apologize for this wrongly written sentence, and corrected as you suggested.

- Page 18: the p=0.0584 does not match with the one in the table.

Corrected 

- Page 19: "an increase of expectation (i.e. one extra partner out of ten) increases the probability (between 0 and 1) to adopt destructive behavior with 0.045." → You mean "a unitary increase in expectation". So, it means that a 10% increase in expectation leads to a 4.5% increase in destruction? right? Maybe it would be more conventional to state it as an elasticity (thus, for a 1% increase). Moreover, can you interpret this? Is it rather inelastic? In a similar vein, check the next sentence as well.

Thank you very much for this interesting suggestion. We present results in terms of percentages.

- Page 21, top: “Both results are in line with Hypothesis 1.” → Why don't you mention that

"Expected partners’ beliefs" is not significant, which contradicts H1? Apparently, you drop this

variable in table 6b but you should explain why. Besides, table 6b is not really introduced in the text. The transition between table 6a and 6b is a bit abrupt.

We now introduce explanatory step by step and results seem to be highly consistent. We also mention the non significance of the variable “Expected partners’ beliefs”.

- Page 22:

o One more time, why using a Tobit regression? You could justify it by specifying the number

of left/right censored observations.

Thank you again for pointing this out. We justified accordingly.

o Typo : “and then regreSS first order beliefs on the second order beliefs (Table 7). To do so,

we use censored Tobit regressions...”

Corrected 

- Page 23: "we should test if there exists an indirect effect of expectations about others’ beliefs on destruction" → the end of the paragraph is strange. You seem to say that you are now going to show that there is an indirect effect, but this is what you just showed in table 7. So, instead of writing “we should test if there exists an indirect effect of expectations about others’ beliefs on destruction. As expected, we found an affirmative response to this assumption.”, I would simply write “This shows that there exists an indirect effect of expectations about others’ beliefs on destruction.”.

We agree and corrected as suggested.

- Page 25:

o "Indeed, experienced participants are closer to real behaviors. " → I think that this is debatable. On the contrary, one might argue that experienced participants have learnt specific behavior that may bias their decisions. For example, they may be used to detect the goal of the experiment. In the positive emotion treatment, they may suspect that the experimenter expects subjects to destroy less, and so be influenced by this conjecture(?)

We thank you for pointing this issue which is indeed a plausible explanation. We included this argument into the manuscript.

o Typo: “and expect that partners belieVE them to be less destructive.” 

Corrected 

o I don’t understand your interpretation regarding the effect of risk aversion. Besides, at the

end of the day, even though you collect ambiguity aversion, you never include it in the regressions (you only focus on risk aversion).

We now also include a variable for ambiguity aversion. However neither of the variables are significant.

o Another global remark on the regressions: you do not consider arousal in the regressions. Is this because the effects are too small? You can simply state it before presenting the regressions.

We don’t use valence or arousal in the regressions. Both variables are used to control for emotion induction (section 5.1)

Discussion & conclusion

- Page 26:

o Typo: “we deduce that individuals in whom positive emotions were induced TO destroy less than others. In other words,” 

We corrected. 

Reviewer #2: 

We would like to thank you again for your careful reading, interesting suggestions and support in helping us to improve our paper. For convenience, we reproduce each of your comments below, followed by our responses in blue.

1.Authors have corrected the inconsistency between “reduce” and “destroy” in the text, but I am not completely sure about the term used in the experiment given that the questions that appeared in the original submissions seemed to be a copy paste from the experiment. Do you authors screen shots of the experiment? 

We provide a screenshot of the experiment (in French) in the supplementary materials

2.Authors use dummies to control for risk and ambiguity aversion. Explain what is considered by authors to be a risk averse and ambiguity averse subject. 

We included some explanations considering the two dummy variables (risk and ambiguity aversion). We consider risk-averse subjects to prefer a certain payoff to a lottery with known probabilities and ambiguity-averse subjects to avoid options whose outcome probabilities are unknown (Ellsberg, 1961). 

3.In table 4, authors state that “destruction is lower in the positive emotion treatment as compared to the negative emotion treatment and the neutral emotion treatment”, but the p-value is 0.78. Then, there are not statistically significant differences between these three treatments. In this table, the differences appear regarding the expectations on partners’ destruction although at a 10% level.

Thank you for pointing this issue. We absolutely agree and modified the discussion of table 4.

4.After Table 5, authors state that “the hypothesis that the proportion of individuals who destroy another player’s money is significantly different under the “positive”, “negative” and “neutral” emotional state is rejected at the 10% level”. The specification of this sentence seems to explain a Kruskal-Wallis test. In this case, there are binary combinations in which the null hypothesis points out towards no differences between the treatments compared. I do not know in which comparison a 10% level appears. In any case, the null hypothesis is always equality of medians, not differences. The result’s explanation is right, but not the previous argument.

We apologize for this wrongly written sentence, and corrected as you suggested.

5.Authors state that “Even if we cannot statistically confirm the tendency that a positive mood reduces the incentive to destroy partner’s endowment, we also observe this tendency for negative emotions contrarily to our hypothesis 2. An explanation that negative emotions (compared to neutral emotions) also decrease incentives to destroy might be explained by the moral compensation phenomenon, or more precisely by moral cleansing” With the data of Table 5 we cannot argue that there are differences between negative and neutral emotions in any of the variables compared.

Again, thank you for pointing this issue. We absolutely agree and modified accordingly. We also mention in the conclusion section that the non-significance of results might also be explained by the effect size.

6.Regarding the ex-post power analysis, what are the value authors have used to calculate it? You present it at the beginning, but the ex-post power analysis needs to be performed for all the test without statistically significant differences. That is, in Table 4 for all the comparison except for the ones with statistically significant results. Moreover, at the beginning you indicate that “a sample size of 50 observations allow us to obtain sufficient power”. I imagine that is the analysis is properly performed you will need 50 observations in each group.

We apologize for not having been more precise on the way we carried out our power analysis. 

We added the following sentences to the manuscript.

We ran an ex-post power analysis using the G*power software which revealed that our sample size (respectively 55, 57 and 60 observations for the three treatments) is sufficient (power=0.8) to observe an effect size of 0.4. While Cohen (1992) recommends that wherein d = 0.20 indicates a small effect, d = 0.50 indicates a medium effect and d = 0.80 indicates a large effect, Dancey & Reidy (2006) specify that small and medium effects are more willingly available when evaluating behavioral and psychological constructs as it is the case of constructs investigated herein. 

7.After result 2, the sentence beginning by “Again” needs to me restructured. It difficult to understand the argument that authors try to convey. I do not see differences regarding negative emotions.

We agree with you, and modified the discussion accordingly. 

8.In table 6b, coefficients are not necessary if you provide marginal effects. Then, you can substitute them and at the end on the table you must specify that values are marginal effects, standard deviations between brackets and the levels of statistically significance. Please, remove coefficients and provide marginal effects for all the variables.

Thank you for this suggestion. We changed tables accordingly and now only provide marginal effects.

9.Given that tables 6a and 6b are measuring the same all the models must be presented in one single table. The first model will have the three explanatory variables included in the reviewed version. The second model will include positive and negative emotion. The third one will include the interactions and the third one will add the control variables. In this case, the models will be adding new variable to see the stability of results. Authors have not controlled for the emotions in the first model, have deleted one main explanatory variable in model 3 and have not controlled for individual characteristics in models 3 and 4. Please, provide the raw data and STATA commands.

Thank you for this suggestion. We now include variables step by step. We also provide raw data and stata commands.

10.At the expense of the changes suggested to the models, the variable “positive emotion” does not refer only to those subjects with low Machiavellian scores. This variable refers to all the subjects in the sample, then, a positive emotions reduced the probability of engaging in destructive behavior. But positive emotions increase destruction of those scoring higher in Machiavellian. Be careful with interpretations and with the sentences about the results.

Thank you for pointing this issue. We have rewritten comments in relation to table 6. 

11.Similarly to my suggestion about the models of Table 6, please run the same models adding variables step by step. Again, provide marginal effects. Please, rewrite the interpretation of positive emotions and the interaction after result 5, it is not clear.

Thank you for this suggestion. We now include variables step by step. We also rewrote the section after result 5 and illustrated results. 

12.Why do not the authors include the variable about first order beliefs in Table 8?

Thank you for this suggestion. Indeed 1st and 2nd order beliefs are highly correlated. We now include the variable about first order beliefs in Table 8.

13.In order to check directly result 6, authors can include in Table 6 an interaction between second order beliefs and emotions.

We choose not to introduce the interaction term between 2nd order beliefs and emotions in table 6. One of the reasons is the correlation between 1st and 2nd order beliefs, which also cancels the significance of 2nd order beliefs in the logit regression. Emotions and 2nd order beliefs are both indirect effects.

14.After Table 8, there is an explanation about table 7. Please, move this explanation after Table 7. I really think that is not accurate to use the term “optimistic”. This result just points out that the fact of having participated in lab experiments previously reduces the expectation of others reducing their partner’s income.

We corrected as suggested.

15.Regarding the explanation about the gender effect, do not use the term “optimistic” and explain it again.

We use the term “optimistic” to explain our result related to the gender effect in relation with findings in the literature.

“We also observe that men have different expectations on partners’ beliefs on one’s own destruction as women: men expect partners to belief them to be less destructive. This result might be explained by findings that men are found to be more optimistic than women [14].”

---

## [Decision Letter · Decision Letter 2]

25 Aug 2022

An experimental investigation on the dark side of emotions and its aftereffects

PONE-D-21-35528R2

Dear Dr. Saadaoui,

We’re pleased to inform you that your manuscript has been judged scientifically suitable for publication and will be formally accepted for publication once it meets all outstanding technical requirements.

Kind regards,

Nikolaos Georgantzis, Dr.

Academic Editor

PLOS ONE

Reviewers' comments:

Reviewer's Responses to Questions

**Comments to the Author**

1. If the authors have adequately addressed your comments raised in a previous round of review and you feel that this manuscript is now acceptable for publication, you may indicate that here to bypass the “Comments to the Author” section, enter your conflict of interest statement in the “Confidential to Editor” section, and submit your "Accept" recommendation.

Reviewer #1: All comments have been addressed

2. Is the manuscript technically sound, and do the data support the conclusions?

Reviewer #1: Yes

3. Has the statistical analysis been performed appropriately and rigorously? 

Reviewer #1: Yes

4. Have the authors made all data underlying the findings in their manuscript fully available?

Reviewer #1: Yes

5. Is the manuscript presented in an intelligible fashion and written in standard English?

Reviewer #1: Yes

6. Review Comments to the Author

Reviewer #1: The authors have worked hard to properly address all my comments. I am now satisfied with the paper.

7. PLOS authors have the option to publish the peer review history of their article (what does this mean?). If published, this will include your full peer review and any attached files.

Reviewer #1: No

---

## [Editor Report · Acceptance letter]

9 Sep 2022

PONE-D-21-35528R2 

An experimental investigation on the dark side of emotions and its aftereffects 

Dear Dr. Saadaoui:

I'm pleased to inform you that your manuscript has been deemed suitable for publication in PLOS ONE. Congratulations! Your manuscript is now with our production department. 

Kind regards, 

on behalf of

Prof. Nikolaos Georgantzis 

Academic Editor

PLOS ONE